# Ecophysiological characteristics of red, green and brown strains of the Baltic picocyanobacterium *Synechococcus* sp. – a laboratory study

S. Śliwińska-Wilczewska[1], A. Cieszyńska[2], and A. Latała[1]

[1]University of Gdańsk, Institute of Oceanography, Laboratory of Marine Plant Ecophysiology, Gdynia, Poland

[2]Institute of Oceanology Polish Academy of Sciences, Department of Marine Physics, Marine Biophysics Laboratory, Sopot, Poland

*Correspondence to:* A. Cieszyńska (acieszynska@iopan.gda.pl, cieszynska.agata@gmail.com)

**Abstract.** The contribution of picocyanobacteria (PCY) to summer phytoplankton blooms, accompanied by an ecological crisis is a new phenomenon in Europe. This issue requires careful investigation. Therefore, the work, which examines the response of *Synechococcus* sp. physiology to different environmental conditions was conducted. Three strains of *Synechococcus* sp. (red BA-120, green BA-124 and brown BA-132) were cultivated in a laboratory under previously determined environmental conditions. These conditions were as follows: temperature (T) from 10 by 5 to 25°C, salinity from 3 by 5 to 18 PSU and Photosynthetically Active Radiation (PAR) from 10 by 90 to 280 µmol photons $m^{-2}$ $s^{-1}$, which gave 64 combinations of synthetic, though real environmental conditions. Scenarios reflecting all possible combinations were applied in the laboratory experiments. Results pointed to differences in final number of cells between strains. However, there was also a similar pattern for BA-124 and BA-132, which showed the highest concentrations of picocyanobacteria cells at higher T and PAR. This was also the case for BA-120, but only to a certain degree as the number of cells started to decrease above 190 µmol photons $m^{-2}$ $s^{-1}$ PAR. Pigmentation, chlorophyll *a* (Chl *a*), fluorescence and rate of photosynthesis presented both similarities and differences between the strains. In this context, more consistent features were observed for brown and red strains when compared to the green. In this paper are defined the ecophysiological responses of PCY.

## 1 Introduction

The presence of picoplankton and its contribution to marine biomass were ignored in environmental studies conducted before 1970. This was related to the poor accuracy of research equipment, which did not enable recording and recognition of such small organisms. Before the discovery of picocyanobacteria (PCY) in the oceans by Johnson and Sieburth (1979) and Waterbury et al. (1979) there only existed incidental reports of this fraction of cyanobacteria occurrence in aquatic ecosystems. Since then, the number of PCY observations has rapidly increased, and currently they are known to be present in many marine, brackish and freshwater ecosystems of the world (e.g., Callieri, 2010; Sorokin and Zakuskina, 2010; Flombaum et al., 2013; Jodłowska and Śliwińska, 2014; Jasser and Callieri, 2017). Additionally, recent works showed that many aquatic ecosystems have been experiencing super-dense, long-term blooms of picocyanobacteria (Sorokin et al., 2004; Sorokin and Zakuskina, 2010), whilst in the past PCY were often described as a non-blooming group (Stockner et al., 1988). Sorokin and Zakuskina (2010) found that the picocyanobacteria blooms were accompanied by great changes in the benthic habitats.

Picocyanobacteria of the *Synechococcus* genus are extremely important organisms in the world's oceans. This is the smallest fraction of plankton ranked by the size of cells, which ranges from 0.2 to 2.0 µm (Sieburth et al., 1978). Chrococcoid genus of the *Synechococcus* are ubiquitous components of the natural plankton communities in aquatic environments. Picocyanobacteria of the *Synechococcus* group span a range of different colors, depending on their pigments

composition (Stomp et al., 2007; Haverkamp et al., 2008). *Synechococcus* sp. ranged by the pigment content are divided into two main groups: strains rich in the pigment phycoerythrin (PE), rendering the representatives a variety of orange, brown, reddish, pink and purple colors, and strains rich in phycocyanin (PC), coloring the organism in various shades of blue-green (Haverkamp et al., 2009). Baltic strains of *Synechococcus* sp. are classified as three main groups: red and brown strains rich in PE and green strains rich in PC (Mazur-Marzec et al., 2013; Jodłowska and Śliwińska, 2014). The difference between red and brown strains is a proportion of two different bilin pigments known as phycoerythrobilin (PEB) and phycourobilin (PUB), which both bind to the PE apoprotein (Everroad and Wood, 2006; Stomp et al., 2007; Six et al., 2007a, b; Haverkamp et al., 2008; 2009). The three strains of *Synechococcus* sp.: BA-120 (red), BA-124 (green), and BA-132 (brown) examined in this work (Fig. S1 in Supplement) are different morphotypes representatives. Coexistence of PE and PC-rich picocyanobacteria can be found in waters of intermediate turbidity, such as many freshwater lakes and coastal seas including the Baltic Sea (Andersson et al., 1996; Hajdu et al., 2007; Stomp et al., 2007; Haverkamp et al., 2008; Haverkamp et al., 2009; Mazur-Marzec et al., 2013; Larsson et al., 2014; Paczkowska et al., 2017).

Picocyanobacterial species are phylogenetically divided into several major clusters. These clusters have been identified, based on photosynthetic pigmentation, nitrogen requirements, motility and salinity preferences (Herdman et al., 2001). Picocyanobacteria that are found and isolated from marine, brackish and freshwater environments are often related to *Synechococcus* cluster 5 (Herdman et al., 2001). *Synechococcus* cluster 5 is divided in two sub-clusters: 5.1 and 5.2. The members of cluster 5.1 typically produce PE as their main photosynthetic pigment. In contrast, members of cluster 5.2 have a green coloration because they produce PC (Herdman et al., 2001; Larsson et al., 2014). The diversity of PCY has been investigated mainly by analysis of the 16S rRNA gene. However, the phylogenetic tree of *Synechococcus* sp. is not always consistent with their pigmentation type (Haverkamp et al., 2008). Thus, the actual taxonomic position may be incorrectly defined due to the morphological plasticity of these organisms (Callieri, 2010).

Despite its association with open ocean systems, it has become increasingly evident in recent years that *Synechococcus* sp. is a significant contributor to cyanobacterial blooms (Beardall, 2008). Surprisingly, this species may also comprise 80% and more of the total cyanobacterial biomass during cyanobacterial blooms in the Baltic Sea (Stal et al., 2003; Mazur-Marzec et al., 2013).

Recently, it has been confirmed that PCY are able to excrete harmful and allelopathic substances (e.g., Jakubowska and Szeląg-Wasilewska, 2015; Jasser and Callieri, 2017; Śliwińska-Wilczewska et al., 2017; Barreiro Felpeto et al., 2018). Many different factors, including physical parameters, availability and competition for resources, selective grazing and allelopathic interactions can affect the occurrence of harmful blooms in aquatic ecosystems. The development of massive algal blooms is a consequence of the interaction between many favorable factors. *Synechococcus* sp. greatly contributes to these massive blooms, but so far the characteristics of the life cycle of Baltic PCY has not been sufficiently studied. This knowledge needs to be expanded and improved, especially because of bloom toxicity and their negative impacts on ecosystems (Jasser and Callieri, 2017; Śliwińska-Wilczewska et al., 2018a).

According to the all above, phytoplankton is of great interest to scientists in terms of understanding its life cycles and impact on the ecosystem in different parts of the world's oceans and within diverse environmental conditions. In order to investigate it, scientists use various types of research methodology: in-situ measurements, laboratory experiments and numerical estimations. All of these approaches are necessary and essential in marine phytoplankton research. Some laboratory and field studies of ecophysiological responses of picocyanobacteria to different growth conditions have already been completed for typical oceanic mediums, semi-closed seas and lakes (e.g., Glover et al., 1986; Kuosa, 1988; Stal et al., 1999; Agawin et al., 2000; Callieri and Stockner, 2002; Hajdu et al., 2007; Sánchez-Baracaldo et al., 2008; Cai and Kong, 2013; Motwani et al., 2013; Jodłowska and Śliwińska, 2014, Stawiarski et al., 2016). However, there is still a need to provide more systematic information about these organisms. What is more, the need is amplified by the fact that there are only a few research papers on the brown strain of Baltic *Synechococcus* sp. (Stal et al., 2003; Haverkamp et al., 2008; 2009; Jodłowska and Śliwińska, 2014). This gives limited knowledge of PCY and their life cycle in the Baltic Sea, as brown form

also contributes to total pico- and phytoplankton biomass in the area of interest (Stal et al., 2003). The above strengthens the motivation to conduct studies on the brown strain of *Synechococcus* sp.

The overall goal of this paper is to determine the most favorable and unfavorable environmental conditions for PCY to grow on the basis of three different strains of *Synechococcus* sp. ecophysiological analysis. What is more, this study aims at describing pigmentation, Chl a fluorescence parameters and photosynthesis performance of PCY cells grown in different environmental conditions. The goal is also to demonstrate how the increasing abundance of PCY in the Baltic Sea may impact the marine ecosystem functioning. The initial step of this study was to carry out laboratory experiments on *Synechococcus* sp. cultures. In order to create different environmental conditions in the Baltic Sea range, combinations of physical quantities were determined. In total, 64 combinations (environmental scenarios) were generated. The second step was to plot and analyze all results after seven days of incubations. For the results, the number of cells, pigmentation, Chl *a* fluorescence parameters, and rate of photosynthesis were collected. The third step was to extract any significant relations between the results and specific physical factors by using a statistical analysis, which included the variance method analysis (two-way ANOVA) and Tukey's HSD post-hoc test. Derived laboratory results help to develop the knowledge on the picocyanobacteria life cycle. Moreover, the PCY experiments underlie the improved numerical approach to phytoplankton modeling development. On the basis of derived results, the algorithms for picocyanobacterium growth is being created in a separate study.

**2 Material and methods**

**2.1 Material and culture conditions**

Three different phenotypes of picocyanobacteria strains from the genus *Synechococcus* were examined: BA-120 (red), BA-124 (green), and BA-132 (brown). The *Synechococcus* sp. strains were isolated from the coastal zone of the Gulf of Gdansk (southern Baltic Sea) and maintained as unialgal cultures in the Culture Collection of Baltic Algae (CCBA) at the Institute of Oceanography, University of Gdańsk, Poland (Latała et al., 2006).

The experiments on the 'batch cultures' were carried out in 25 mL glass Erlenmeyer flasks containing sterilized f/2 medium (Guillard, 1975). Culture media was prepared with artificial seawater filtered through a 0.45-µm filters (Macherey-Nagel MN GF-5) using a vacuum pump (600 mbar) and autoclaved. The cultures were incubated in 35 mL Erlenmeyer glass flasks. Salinity of the media was prepared by dissolving Tropic Marine Synthetic Sea Salt in distilled water. The major nutrients, microelements and vitamin concentrations were added according to a method proposed by Guillard (1975) (any of the components in f/2 media were not replaced by Tropic Marine Synthetic Sea Salt).

The PCY cultures were adapted to the various synthetic environmental conditions for two days. The conditions were the combinations of different values of: scalar irradiance in Photosynthetically Active Radiation (PAR) spectrum (10, 100, 190 and 280 µmol photons $m^{-2}$ $s^{-1}$), temperature (T) (10, 15, 20 and 25°C), and salinity (3, 8, 13 and 18 PSU). The salinity was controlled by salinometer (inoLab Cond Level 1, Weilheim in Oberbayern, Germany). The intensity of PAR was measured using a LI-COR spherical quantum-meter (LI-189, LI-COR Inc., Nebraska, USA). Fluorescent lamps (Cool White 40W, Sylvania, USA) were used as source of irradiance and combined with halogen lamps (100W, Sylvania, USA) to obtain more intensive light. Both light sources give PAR spectrum. This was proved by Jodłowska and Latała (2010) and Jodłowska and Śliwińska (2014). What is more, LI-COR manual with technical specification therein, says that the sensor first checks the light spectrum and if it responds PAR spectrum, the intensity of radiation is measured. This implies, all the results given by LI-COR refers to PAR. Values of quantities representing each environmental condition were applied at the fixed intervals, i.e.: PAR, interval 90; T, interval 5; salinity, interval 5.

The synthetic environmental conditions of salinity and T applied in the laboratory are representative for the Baltic Sea area (Feistelet al., 2008; 2009; Siegel and Gerth, 2017). Moreover, the values of environmental conditions variables (salinity,

temperature, PAR) were also specified in certain ranges to make this study comparable with other laboratory cultures experiments available in literature. The combination of the quantities of environmental variables is called a scenario in the present paper. After acclimation time (2 d), the PCY cells served as inoculum for the right test cultures with the initial number of cells equal to $10^6$ cells mL$^{-1}$. The flasks with picocyanobacteria were shaken (once a day) during the experiment. In order to achieve the most reliable results, test cultures were grown in three replicas and were incubated for one week at each combination of light, temperature and salinity. On the last day of incubation the number of cells, pigment content, Chl $a$ fluorescence, and rate of photosynthesis were measured in each replica. Results were reported as mean values ± standard deviation (SD).

**2.2 Determination of the number of cells**

The flow cytometry was used to establish the initial number of picocyanobacteria cells and to measure the final cells concentration after the incubation period. The number of cells (N) in cultures was counted with flow cytometer BD Accuri™ C6 Plus (BD Biosciences, San Jose, CA, USA) according to the procedure proposed by Śliwińska-Wilczewska et al. (2018b). Events were recorded in list form. Samples were run at a flow rate of approximately 14 µL min$^{-1}$. Selection of this flow rate was based on previous introductory experiments to determine the most relevant effectiveness. Choosing an adequate discriminator and thresholds plays a key role in recording the cells correctly. The most reasonable solution to record chlorophyll fluorescing cyanobacteria and microalgae is to choose the red fluorescence as the discriminator (Fig. S1) and to select a high threshold, high enough to eliminate optical and electronic noise (Marie et al., 2005). Concerning this, the discriminator was set on the red (chlorophyll) fluorescence with a standard threshold of 80,000 on FSC-H. Flow was daily calibrated with Spherotech 6- and 8- Peak Validation Beads (BD, San Jose, USA). This ensures that the cytometer works properly and is accurately calibrated for running experiments. Fluorescein isothiocyanate (FITC), phycoerythrin (PE), and PE-Cy5 detectors were daily calibrated with SPHERO™ Rainbow Calibration Particles (BD, San Jose, USA), and the Allophycocyanin (APC) channel was calibrated with SPHERO 6-peaks Allophycocyanin Calibration Particles. Detectors FL1, FL2, and FL3 read fluorescence emissions excited by the blue laser (480 nm), while detector FL4 reads emissions excited by the red laser (640 nm).

**2.3 Determination of the pigments content**

The concentration of photosynthetic pigments of analyzed picocyanobacteria was measured by the spectrophotometric method (Strickland and Parsons, 1972). The analysis of mL-specific (pigment content per mL) and cell-specific (pigment content per cell) pigmentation was conducted. Note that mL-specific means volume-specific, whereas the volume is fixed to 1 mL. After seven days of incubation, 4 mL of culture was filtered in order to separate the picocyanobacteria cells from the medium. Chl $a$ and carotenoids (Car) were extracted from the PCY cells with cold 90% acetone (5 mL). To improve extraction, the cells were disintegrated for two minutes by ultrasonication. Then, the test-tube with the extract was held in the dark for three hours at -60°C. To remove cell debris and filter out the particles, the extracts were centrifuged at 10,000 rpm ($8496 \times g$) for 5 min (Sigma 2-16P, Osterode am Harz, Germany). The absorbance of pigments was estimated on the basis of Beckman spectrophotometer UV-VIS DU 530 measurements at specific wavelengths (750, 665 and 480 nm), using 1 cm quartz cuvette. Pigment concentration was calculated according to Strickland and Parsons (1972). The following formulas have been used: Chl $a$ (µg mL$^{-1}$) = $11.236(A_{665}\text{-}A_{750})V_a/V_b$, Car (µg mL$^{-1}$) = $4(A_{480}\text{-}A_{750})V_a/V_b$, where: $V_a$ - extract volume (in this study 5 mL), $V_b$ - sample volume (in this study 4 mL), and $A_x$ - absorbance estimated at wavelength x in a 1-cm cuvette.

**2.4 Chlorophyll fluorescence analyses**

Chl *a* fluorescence was measured with a Pulse Amplitude Modulation (PAM) fluorometer (FMS1, Hansatech, King's Lynn, Norfolk, UK). The FMS1 uses a 594 nm amber modulating beam with 4-step frequency control as a measuring light and is equipped with a dual-purpose halogen light source providing actinic light (0 – 3000 µmol photons m$^{-2}$ s$^{-1}$ in 50 steps) and a saturating pulse (0 – 20000 µmol photons m$^{-2}$ s$^{-1}$ in 100 steps). FMS1 also has a 735 nm far-red LED source for preferential PSI excitation allowing accurate determination of the $F_o$' parameter. Samples were filtered onto 13-mm glass fiber filters (Whatman GF/C, pore size = 1.2 µm). Before measurement, the filtered sample was kept in the dark for 10 min. The maximum photochemical efficiency of photosystem II (PSII) at dark-adapted state ($F_v/F_m$) and the photochemical efficiency of PSII under actinic light intensity (ΦPSII) were estimated. The actinic light was different for cultures grown in different environmental conditions and referred to the PAR value in respective scenarios. The above is similar to the method used by Campbell et al. (1998).

**2.5 Measurements of photosynthesis rate**

The measurements of oxygen evolution were carried out on the seventh day of the experiment using a Clark-type oxygen electrode (Chlorolab 2, Hansatech). Temperature was controlled with a cooling system LAUDA (E100, Germany). Illumination was provided by a high intensity probe-type light array with 11 red LED's centered on 650 nm. Irradiance was measured with a quantum sensor (Quantitherm, Hansatech, King's Lynn, Norfolk, UK). Dark respiration was estimated from $O_2$ uptake by cells incubated in the dark. Experimental data (photosynthetic parameters, i.e., the photosynthetic capacity ($P_m$), the initial slope of *P-E* curve ($\alpha$) and the dark respiration ($R_d$)) was fitted to the photosynthetesis irradiance response (*P-E*) curves using equation (Jassby and Platt, 1976) and Statistica® 13.1 software (Sakshaug et al., 1997).

**2.6 Statistical analyses**

The effect of light and temperature separately and then their combinations impact on growth, pigments content, fluorescence and photosynthesis performance of examined strains were analyzed using two-way variance analysis (ANOVA). A post hoc test (Tukey's HSD) was used to show which results differed under varied conditions over the experimental period (Sheskin 2000). The confident levels in the statistical analysis were: 95% (*$p < 0.05$), 99% (**$p < 0.01$), 99.9% (***$p < 0.001$). The statistical analyses were performed using Statistica® 13.1 and Matlab 2012b software. According to the literature, light and temperature are major factors controlling the growth and distribution of picocyanobacteria (e.g.: Jasser and Arvola, 2003), and they may have considerable significance on the abundance of the *Synechococcus* community (Glover, 1985; Glover et al., 1985; 1986, Joint and Pomroy, 1986; Jasser and Arvola, 2003; Jasser, 2006; Jodłowska and Śliwińska, 2014). Due to that, it was decided that light and temperature would be the independent variables in ANOVA and post-hoc test analysis. The dependent variable was always the parameter, which had been measured.

**3 Results**

**3.1 Number of cells**

For all three picoplankton strains, ANOVA analysis indicated that in each scenario the independent variable (temperature or PAR) significantly influenced the dependent variable. What is more, post-hoc tests indicated that multiple factors (T and PAR together) had an impact on the PCY growth.

According to post-hoc tests, 2008 multiple comparisons (70%) out of all 2880 completed for three strains, indicated the highest statistical significance (Tukey HSD, *** $p < 0.001$), 160 multiple comparisons (6%) pointed to the statistical

significance of $0.001 < ** \, p < 0.01$, and 114 (4%) showed the significance of $0.01 < * \, p < 0.05$. The rest of the multiple comparisons (598, 20%) indicated no statistically significance differences (Tukey HSD, $p \geq 0.05$).

Both PAR and T affected the number of *Synechococcus* sp. BA-120 cells significantly *(ANOVA, $F_{9,32} = 42.3$, *** $p < 0.001$, ANOVA, $F_{9,32} = 22.7$, *** $p < 0.001$, ANOVA, $F_{9,32} = 9.6$, *** $p < 0.001$ and ANOVA, $F_{9,32} = 12.2$, *** $p < 0.001$, for salinity 3, 8, 13, 18 PSU, respectively)*. For BA-120, the number of cells increased with T in each medium (salinities 3, 8, 13, 18 PSU) (Fig. 1A, a-d). The minimum number of cells was estimated in salinity 3 PSU, T 10°C and PAR 10 μmol photons m$^{-2}$ s$^{-1}$ ($1.6\times10^6$ cell mL$^{-1}$, Fig. 1A, a), whilst the maximum in salinity 18 PSU, T 25°C, PAR 190 μmol photons m$^{-2}$ s$^{-1}$ ($11.5\times10^6$ cell mL$^{-1}$, Fig. 1A, d). The decrease in number of cells was observed from PAR 190 μmol photons m$^{-2}$ s$^{-1}$ onwards. This can likely be related to the photo-inhibition of photosystem II (PSII). The above was the case in each salinity (Figs. 1A, a-d). Additionally, the results analysis (Fig 1A, a-d) showed that the most important environmental factor influencing BA-120 number of cells was T, with PAR playing an additional role, for instance in the context of photo-inhibition. This was pronounced the most within lower temperatures (10 and 15°C), where the change in BA-120 abundance along with PAR increase was barely observed being plainly visible along with T increase at once. Multiple comparisons tests pointed to the strong significance of PAR and T combined in influencing the number of *Synechococcus* sp. BA-120 cells. According to the statistics, 82% of multiple comparisons were statistically significant (Tukey HSD, * $p < 0.05$) with 91% of them having the highest significance level (Tukey HSD, *** $p < 0.001$).

Both PAR and T also significantly affected the number of *Synechococcus* sp. BA-124 cells *(ANOVA, $F_{9,32} = 7.9$, *** $p < 0.001$, ANOVA, $F_{9,32} = 13.6$, *** $p < 0.001$, ANOVA, $F_{9,32} = 8.4$, *** $p < 0.001$ and ANOVA, $F_{9,32} = 2.8$, ** $p < 0.01$, for salinity 3, 8, 13, 18 PSU, respectively)*. For BA-124, number of cells increased with T and PAR in all salinities (Figs. 1B, a-d). The lowest number of cells was calculated in salinity 3 PSU, T 10°C and PAR 10 μmol photons m$^{-2}$ s$^{-1}$ ($2.0\times10^6$ cell mL$^{-1}$, Fig. 1B, a) and the highest number of cells was reached in salinity 18 PSU, T 25°C, PAR 280 μmol photons m$^{-2}$ s$^{-1}$ ($43.6\times10^6$ cell mL$^{-1}$, Fig. 1B, d). High abundances were estimated also under the highest T and PAR conditions in salinity 13 PSU, where a number of cells equalled $41.1\times10^6$ cell mL$^{-1}$ (Fig. 1B, c). Generally, the number of cells was the highest in BA-124 cultures when compared to BA-120 and BA-132 cultures in respective scenarios. One of the observations was the difference in BA-124 number of cells between lower and higher PAR and T conditions (scenarios with lower PAR and T and scenarios with higher PAR and T). BA-124 seemed to be more sensitive to changes in PAR and T in their lower rather than in higher ranges. Regarding salinity, the highest number of BA-124 cells were noted in moderate- and high-salinity mediums. Optimum salinities for strain BA-124 were 8 and 13 PSU. Due to post-hoc analysis, salinity 13 PSU differentiated the conditions for cell abundances under different PAR and T at a lower degree when compared to other salinities under respective PAR and T (the least statistically significant differences observed in medium 13 PSU), which is also noticeable in Fig. 1B, c. Another feature of BA-124 was the number of cells in low T and high PAR scenarios were nearly equal to cell abundances in high T and low PAR scenarios. This was not the case for BA-120 and BA-132 strains. The observation was supported by Tukey's tests, where only few statistically significant differences in number of cells were observed between scenarios with elevated PAR (280 μmol photons m$^{-2}$ s$^{-1}$), low T (10, 15°C) and those with high T (25°C) and low PAR (10 μmol photons m$^{-2}$ s$^{-1}$). These differences were observed between 15°C and 280 μmol photons m$^{-2}$ s$^{-1}$ and 25°C and 10 μmol photons m$^{-2}$ s$^{-1}$ in salinities 3 and 8 PSU (Tukey HSD, ** $p < 0.05$ in both cases, Figs. 1B, a-b). Multiple comparisons tests showed high significance of combinations of PAR and T in affecting the number of cells. According to Tukey HSD tests, 72% of multiple comparisons were statistically significant (* $p < 0.05$) with 82% of them with the highest significance level (*** $p < 0.001$).

Similarly to BA-120 and BA-124, it was found that PAR and T significantly affected the number of *Synechococcus* sp. BA-132 cells *(ANOVA, $F_{9,32} = 6.8$, *** $p < 0.001$, ANOVA, $F_{9,32} = 5.4$, *** $p < 0.001$, ANOVA, $F_{9,32} = 5.6$, *** $p < 0.001$ and ANOVA, $F_{9,32} = 12.5$, ** $p < 0.01$, for salinity 3, 8, 13, 18 PSU, respectively)*. For BA-132, the positive impact of T and PAR on number of cells (Figs. 1C, a-d) was observed in each medium. Note that positive impact means the increasing (positive) dependency, whilst negative impact means decreasing (negative) dependency between the independent and

dependent variable, e.g.: between T and abundance. Salinity played a more significant role here than when compared to BA-124. It was found that the higher the salinity, the higher the number of cells of BA-132. What is more, according to the statistical analysis, salinity 18 PSU differentiated the number of cells the most (Fig. 1C, d). In salinity 18 PSU, the cell abundances could be described as a linear increasing function of ambient T and PAR. This was also observed in other salinities but not as intensively pronounced as in the highest-saline medium. Moreover, in high salinity, the sensitivity of number of cells to T changes was much lower than in low salinities. PAR did not determine the number of cells as strongly as T, which was quite consistent to the observation noted for BA-120. The minimum number of cells was observed in 3 PSU, 10°C and 10 µmol photons $m^{-2} s^{-1}$ ($1.4 \times 10^6$ cell $mL^{-1}$, Fig. 1C, a), whilst the maximum in 18 PSU, 25°C, 280 µmol photons $m^{-2} s^{-1}$ ($16.1 \times 10^6$ cell $mL^{-1}$, Fig. 1C, d). In addition, the lowest values of BA-132 number of cells were calculated for the lowest T and PAR condition in each salinity. Tukey HSD post hoc tests indicated high significance of the combination of PAR and T in affecting the cell abundances. Regarding those tests, 84% of multiple comparisons were statistically significant (* $p < 0.05$) with 90% of them with the highest significance (*** $p < 0.001$).

Concerning all three strains, high salinity generally had a positive impact on number of *Synechococcus* sp. cells. What is more, the relations between salinity and number of cells for all strains, especially red and brown were almost increasing linearly with the highest average increase for BA-132.

**3.2 Pigment content**

The results showed that for all strains, cell-specific pigment composition (pigment content per cell) was environmentally driven (Figs. 2, 3). The analysis of mL-specific pigmentation (pigment content per mL) was also done (Figs. S2 and S3 in Supplement), however, the mL-specific pigment content is another way to illustrate the biomass and that is why it is not described in this section in detail.

It was estimated, that PAR and T significantly affected the Chl *a* cell-specific content of *Synechococcus* sp. BA-120 *(ANOVA, $F_{9,32} = 33.7$, *** $p < 0.001$, ANOVA, $F_{9,32} = 5.3$, *** $p < 0.001$, ANOVA, $F_{9,32} = 15.6$, *** $p < 0.001$ and ANOVA, $F_{9,32} = 5.7$, *** $p < 0.001$, for salinity 3, 8, 13, 18 PSU, respectively)*. Both PAR and T also affected the Car content in the BA-120 strain cells significantly *(ANOVA, $F_{9,32} = 25.8$, *** $p < 0.001$, ANOVA, $F_{9,32} = 7.5$, *** $p < 0.001$, ANOVA, $F_{9,32} = 7.3$, *** $p < 0.001$, and ANOVA, $F_{9,32} = 12.0$, *** $p < 0.001$, for salinity 3, 8, 13, 18 PSU, respectively)*. It was found that cell-specific Chl *a* and Car concentrations decreased with the increase of salinity (Figs. 2A, 3A). On average, the cell content of pigments for BA-120 was the highest when compared to the other strains. Chl *a* concentration dominated over Car concentration in each scenario. What is more, there were very high cell-specific concentrations of Chl *a* observed for the whole T range at low PAR. Maximum Chl *a* content was measured under T 25°C and PAR 10 µmol photons $m^{-2} s^{-1}$. This was the case in each salinity. The highest Chl *a* concentration within all scenarios was reached in BA-120 cells in salinity 3 PSU and was equal to 0.339 pg $cell^{-1}$ (Fig. 2A, a). For other salinities these maximums were as follows: 0.233 pg $cell^{-1}$ (8 PSU, Fig. 2A, b), 0.164 pg $cell^{-1}$ (13 PSU, Fig. 2A, c), 0.100 pg $cell^{-1}$ (18 PSU, Fig. 2A, d). The highest Car content was measured in salinity 3 PSU under T of 20°C and PAR 10 µmol photons $m^{-2} s^{-1}$ and reached 0.160 pg $cell^{-1}$ (Fig. 3A, a). The lowest concentrations of Chl *a* (0.038 pg $cell^{-1}$) and Car (0.031 pg $cell^{-1}$) were measured in salinity 18 PSU, T 25°C, PAR 190 µmol photons $m^{-2} s^{-1}$ (Fig. 2A, d) and salinity 18 PSU, T 15°C, PAR 280 µmol photons $m^{-2} s^{-1}$ (Fig. 3A, d), respectively. Multiple comparisons tests indicated the significance of PAR and T combined in shaping the pigmentation. Due to those tests, 52% and 55% of multiple comparisons in Chl *a* and Car content analysis, respectively, were statistically significant (Tukey HSD, * $p < 0.05$) with 80% (for Chl *a*) and 74% (for Car) of them with the highest significance (Tukey HSD, *** $p < 0.001$).

Both PAR and T affected the Chl *a* cell-specific content *(ANOVA, $F_{9,32} = 3.3$, ** $p < 0.01$, ANOVA, $F_{9,32} = 8.3$, *** $p < 0.001$, ANOVA, $F_{9,32} = 69.8$, *** $p < 0.001$ and ANOVA, $F_{9,32} = 17.5$, *** $p < 0.001$, for salinity 3, 8, 13, 18 PSU, respectively)* and Car cell-specific content *(ANOVA, $F_{9,32} = 4.6$, *** $p < 0.001$, ANOVA, $F_{9,32} = 65.5$, *** $p < 0.001$,*

*ANOVA, $F_{9,32} = 83.1$, \*\*\* p < 0.001 and ANOVA, $F_{9,32} = 43.2$, \*\*\* p < 0.001, for salinity 3, 8, 13, 18 PSU, respectively)* of
*Synechococcus* sp. BA-124 significantly. Generally, PAR and high T increase had a negative impact on pigmentation (Figs.
2B, 3B). Maximum values of cell-specific Chl *a* and Car concentrations were measured under 10°C and 10 μmol photons m$^{-}$
$^2$ s$^{-1}$ in each salinity medium. These values, concerning salinities from the lowest to the highest, were as follows: 0.095,
0.102. 0.176, 0.148 pg cell$^{-1}$ for Chl *a* (Figs. 2B, a-d) and 0.051, 0.067, 0.087, 0.079 pg cell$^{-1}$ for Car (Figs. 3B, a-d).
Nonetheless, there were also some exceptions. In salinity 3 PSU, high Car contents were calculated under 280 μmol photons
m$^{-2}$ s$^{-1}$ and T: 15, 20°C and equaled to 0.042 pg cell$^{-1}$ and 0.041 pg cell$^{-1}$, respectively (Fig. 3B, a). On average, salinity
increase had a negative impact on pigmentation. The lowest cell-specific concentrations of Chl *a* and Car in BA-124 cells
were estimated in the same scenario: salinity 18 PSU, T 10°C, PAR 280 μmol photons m$^{-2}$ s$^{-1}$ and were equal to 0.013 pg
cell$^{-1}$ (Fig. 2B, d) and 0.009 pg cell$^{-1}$ (Fig. 3B, d), for Chl *a* and Car, respectively. Multiple comparisons tests pointed to the
significance of PAR and T combined in influencing the pigmentation. According to the statistics, 47% and 54% of multiple
comparisons in Chl *a* and Car content analysis, were statistically significant (Tukey HSD, \* p < 0.05) with 83% (for Chl *a*)
and 79% (for Car) of them with the highest significance level (Tukey HSD, \*\*\* p < 0.001).

It was also examined that PAR and T affected the Chl *a* cell-specific content *(ANOVA, $F_{9,32} = 6.5$, p < 0.001, ANOVA,*

*$F_{9,32} = 11.1$, p < 0.001, ANOVA, $F_{9,32} = 21.5$, p < 0.001 and ANOVA, $F_{9,32} = 6.5$, p < 0.001, for salinity 3, 8, 13, 18 PSU,*
*respectively)* and Car cell-specific content *(ANOVA, $F_{9,32} = 8.6$, p < 0.001, ANOVA, $F_{9,32} = 9.6$, p < 0.001, ANOVA, $F_{9,32} =$*
*4.6, p < 0.001 and ANOVA, $F_{9,32} = 26.8$, p < 0.001, for salinity 3, 8, 13, 18 PSU, respectively)* of *Synechococcus* sp. BA-132
significantly. It was found that salinity increase had a negative impact on cell-specific Chl *a* and Car concentrations. BA-132
was richer in cell-specific pigments than BA-124 (Figs. 2C, 3C). Along with PAR increase, the Chl *a* concentration
decreased significantly. The maximum Chl *a* cell-specific content was measured in moderate or high T (20°C in salinity 13
PSU and 25°C in salinity 3, 8, 18 PSU) under the lowest PAR (10 μmol photons m$^{-2}$ s$^{-1}$). These maximums were 0.299 pg
cell$^{-1}$ in salinity 3 PSU (Fig. 2C, a), 0.248 pg cell$^{-1}$ in salinity 8 PSU (Fig. 2C, b), 0.151 pg cell$^{-1}$ in salinity 13 PSU (Fig. 2C,
c) and 0.073 pg cell$^{-1}$ in salinity 18 PSU (Fig. 2C, d). Consistently with Chl *a*, Car cell-specific content maximums were also
measured under the lowest PAR (10 μmol photons m$^{-2}$ s$^{-1}$) but contrary to Chl *a*, at the lowest T (10°C). These maximums
were: 0.194 pg cell$^{-1}$ in salinity 3 PSU (Fig. 3C, a), 0.131 pg cell$^{-1}$ in salinity 8 PSU (Fig. 3C, b), 0.097 pg cell$^{-1}$ in salinity 13
PSU (Fig. 3C, c), 0.062 pg cell$^{-1}$ in salinity 18 PSU (Fig. 3C, d). Minimums of Chl *a* and Car cell-specific contents within all
scenarios were estimated in salinity 18 PSU, T 15°C and PAR 280 μmol photons m$^{-2}$ s$^{-1}$ being equal to 0.020 pg cell$^{-1}$ (Fig.
2C, d) and 0.19 pg cell$^{-1}$ (Fig. 3C, d), for Chl *a* and Car, respectively. Regarding Chl *a* for minimum content per cell the
same concentration as above mentioned (0.020 pg cell$^{-1}$) was also estimated in salinity 13 PSU for the same conditions of T
and PAR (Fig. 2C, c). Tukey HSD tests pointed to the significance of PAR and T combined in impacting the pigmentation.
According to those tests, 66% and 61% of multiple comparisons in Chl *a* and Car content analysis, respectively, were
statistically significant (Tukey HSD, \* *p* < 0.05), with 81% (for Chl *a*) and 75% (for Car) of them with the highest
significance (Tukey HSD, \*\*\* p < 0.001).
**3.3 Chl *a* fluorescence**
The parameters of Chl *a* fluorescence were depicted as two-factor-dependent graphs, where the values in between the
specific measurements were interpolated (Figs. 4, 5). For all strains, Chl *a* fluorescence parameters were measured and
examined. These parameters were: the maximum photochemical efficiency of photosystem II (PSII) at dark-adapted state
($F_v/F_m$) and the photochemical efficiency of PSII under actinic light intensity (ΦPSII).

The results showed that PAR and T affected $F_v/F_m$ *(ANOVA, $F_{9,32} = 5.2$, p < 0.001, ANOVA, $F_{9,32} = 5.7$, p < 0.001,*

*ANOVA, $F_{9,32} = 4.8$, p < 0.001 and ANOVA, $F_{9,32} = 33.9$, p < 0.001, for salinity 3, 8, 13, 18 PSU, respectively)* and ΦPSII
*(ANOVA, $F_{9,32} = 4.5$, p < 0.001, ANOVA, $F_{9,32} = 5.7$, p < 0.001, ANOVA, $F_{9,32} = 6.3$, p < 0.001 and ANOVA, $F_{9,32} = 2.3$, p <*
*0.05, for salinity 3, 8, 13, 18 PSU, respectively)* of *Synechococcus* sp. BA-120 significantly. For this strain, especially in low
T scenarios and in all scenarios with the lowest salinity, higher $F_v/F_m$ was observed for 280 μmol photons m$^{-2}$ s$^{-1}$ when
compared to 190 μmol photons m$^{-2}$ s$^{-1}$ (Fig. 4A, a). Generally, strong fluctuations were noticeable in $F_v/F_m$ values, which
disabled the fixed environmentally driven pattern determination. However, there was a constant relation noted between T
and PAR and ΦPSII. PAR and T increase had a negative impact on ΦPSII. The impact was the strongest in low salinity
(Figs. 5A, a-b). Nonetheless, in each salinity, the lowest ΦPSII were observed under the highest T and elevated PAR (190 or
280 μmol photons m$^{-2}$ s$^{-1}$). On the contrary, the highest ΦPSII values were calculated in the lowest T and PAR conditions in
every salinity. The highest $F_v/F_m$, for all BA-120 experiments equaled 0.804 and was estimated for scenario: salinity 18
PSU, T 10°C, PAR 280 μmol photons m$^{-2}$ s$^{-1}$ (Fig. 4A, d). Generally, maximum values of $F_v/F_m$ in each medium were
associated with the lowest temperature. Minimum $F_v/F_m$ within all scenarios was estimated for salinity 3 PSU, T 25°C and
PAR 190 μmol photons m$^{-2}$ s$^{-1}$ (0.409, Fig. 4A, a). Concerning ΦPSII, the greatest value was 0.768 estimated in salinity 18
PSU, T 10°C and PAR 10 μmol photons m$^{-2}$ s$^{-1}$ (Fig. 5A, d). Minimum ΦPSII was measured in salinity 3 PSU, T 25°C and
PAR 280 μmol photons m$^{-2}$ s$^{-1}$ (0.241, Fig. 5A, a). Multiple comparisons tests pointed to a strong environmental influence
on Chl *a* fluorescence parameters. Regarding $F_v/F_m$, 65% of all comparisons were statistically significant (Tukey HSD, * $p <$
0.05) with 78% of them having the highest significance (Tukey, HSD, *** $p < 0.001$). For ΦPSII the percentages were as
follows: 80% of all comparisons were statistically significant (Tukey HSD, * $p < 0.05$) and 87% of them had the highest
significance (*** $p < 0.001$).
Both PAR and T significantly affected $F_v/F_m$ *(ANOVA, $F_{9,32} = 46.2$, *** $p < 0.001$, ANOVA, $F_{9,32} = 5.1$, *** $p < 0.001$,*
*ANOVA, $F_{9,32} = 5.0$, *** $p < 0.001$ and ANOVA, $F_{9,32} = 20.6$, *** $p < 0.001$, for 3, 8, 13, 18 PSU, respectively)* and ΦPSII
*(ANOVA, $F_{9,32} = 25.0$, *** $p < 0.001$, ANOVA, $F_{9,32} = 11.6$, *** $p < 0.001$, ANOVA, $F_{9,32} = 15.4$, $p < 0.001$ and ANOVA,*
*$F_{9,32} = 5.2$, $p < 0.001$, for 3, 8, 13, 18 PSU, respectively)* of *Synechococcus* sp. BA-124. For this strain, $F_v/F_m$ reached the
lowest values when compared to the respective incubations of other strains. The values of $F_v/F_m$ generally decreased along
with PAR and T increases but with some exceptions. Generally, ΦPSII environmentally driven characteristics were similar to
$F_v/F_m$ characteristics. The $F_v/F_m$ minimums were measured under the lowest T and highest PAR in each salinity (Figs. 4B, a-
d). The lowest value within all scenarios was 0.124 and was observed in salinity 3 PSU, T 10°C and PAR 280 μmol photons
m$^{-2}$ s$^{-1}$ (Fig. 4B, a). The $F_v/F_m$ maximums were estimated for the highest T and the lowest PAR in each salinity. The highest
$F_v/F_m$ equaled 0.560 for salinity 3 PSU, T 25°C and PAR 10 μmol photons m$^{-2}$ s$^{-1}$ (Fig. 4B, a). Minimums of ΦPSII,
consistently with $F_v/F_m$, were noted under the lowest T and highest PAR. The lowest ΦPSII within all BA-124 experiments
was 0.114 (followed by the minimum in salinity 3 PSU being equal to 0.116, Fig. 5B, a) and was measured in salinity 13
PSU (Fig. 5B, c). Maximums of ΦPSII were observed in the highest T and lowest PAR in each medium, similarly to $F_v/F_m$.
The greatest value of ΦPSII was 0.542 and was measured in salinity 3 PSU, T 25°C and PAR 10 μmol photons m$^{-2}$ s$^{-1}$ (Fig.
5B, a). Tukey HSD post hoc test showed that PAR and T combined influenced Chl *a* fluorescence parameters significantly.
Concerning $F_v/F_m$, 77% of all comparisons were statistically significant (* $p < 0.05$) with 88% of them having the highest
significance (*** $p < 0.001$). For ΦPSII the percentages were as follows: 79% of all comparisons were statistically
significant (* $p < 0.05$) and 89% of them had the highest significance (*** $p < 0.001$).
It was found that both PAR and T affected $F_v/F_m$ *(ANOVA, $F_{9,32} = 4.3$, $p < 0.001$, ANOVA, $F_{9,32} = 4.8$, $p < 0.001$,*
*ANOVA, $F_{9,32} = 4.5$, $p < 0.001$ and ANOVA, $F_{9,32} = 5.7$, $p < 0.001$, for salinity 3, 8, 13, 18 PSU, respectively)* and ΦPSII
*(ANOVA, $F_{9,32} = 10.1$, $p < 0.001$, ANOVA, $F_{9,32} = 7.7$, $p < 0.001$, ANOVA, $F_{9,32} = 4.7$, $p < 0.001$ and ANOVA, $F_{9,32} = 7.0$, $p*
*< 0.001$, for salinity 3, 8, 13, 18 PSU, respectively)* of *Synechococcus* sp. BA-132, significantly. For this strain, $F_v/F_m$
decreased along with the PAR increase but was positively affected by T in each salinity (Figs. 4C, a-d). Minimum values of
$F_v/F_m$ were measured in the highest PAR and the lowest T in each salinity. The lowest $F_v/F_m$ within all experiments on BA-
132 was estimated in salinity 13 PSU ($F_v/F_m$ = 0.155, Fig. 4C, c). In salinity 3 PSU, under aforementioned conditions of T
and PAR, the $F_v/F_m$ value was also low compared to the others and equaled 0.160 (Fig. 4C, a). The maximums of $F_v/F_m$
were measured in T 25°C and PAR 10 μmol photons m$^{-2}$ s$^{-1}$. This was the case for all mediums. The highest $F_v/F_m$ were
noted in salinities 13 and 18 PSU and equaled 0.742 (Fig. 4C, c) and 0.733 (Fig. 4C, d), respectively. The lowest ΦPSII were

noted under the highest PAR and T conditions in every salinity (Figs. 5C, a-d). The minimum $\Phi$PSII, within all gathered results, was obtained in salinity 3 PSU and equaled 0.281 (Fig. 5C, a). Maximums of $\Phi$PSII were measured under completely opposite conditions to the ones stating for minimums, i.e. the lowest PAR and T. The highest $\Phi$PSII, 0.786, was noted in salinity 8 PSU, T 10°C and PAR 10 µmol photons $\mathrm{m}^{-2}\,\mathrm{s}^{-1}$ (Fig. 5C, b). The $\Phi$PSII reached generally higher values than $F_\mathrm{v}/F_\mathrm{m}$ in BA-132 experiments. $\Phi$PSII reached lower values than $\Phi$PSII measured under respective conditions for two other strains. Multiple comparisons tests point to a strong environmental influence on Chl $a$ fluorescence parameters. For $F_\mathrm{v}/F_\mathrm{m}$, 78% of all comparisons were statistically significant (Tukey HSD, * $p < 0.05$) with 89% of them with the highest significance (Tukey, HSD, *** $p < 0.001$). For $\Phi$PSII, 82% of all comparisons were statistically significant (Tukey HSD, * $p < 0.05$), with 89% of them having the highest significance level (Tukey, HSD, *** $p < 0.001$).

Generally, for the BA-120 strain, $F_\mathrm{v}/F_\mathrm{m}$ was affected negatively by T increase, while BA-124 and BA-132 strains were affected positively. T increase had a positive impact on $\Phi$PSII for BA-124 and a negative impact for BA-120 and BA-132. On average, $\Phi$PSII decreased along with PAR increase in all cultures.

**3.4 Photosynthesis**

Net photosynthetic light-response curves for three PCY strains were analyzed. For all cultures, the photosynthesis parameters were: maximum of photosynthesis, photosynthesis efficiency at low irradiance, and dark respiration ($P_\mathrm{m}$, $\alpha$, $R_\mathrm{d}$, respectively) and these were estimated for Chl $a$-specific and cell-specific domains (Figs. S4-S6 in Supplement). It should be noted that dark respiration values were negative (less oxygen than carbon dioxide ($CO_2$)), which meant the lower $R_\mathrm{d}$, the less net oxygen concentration was. This, in turn, indicated higher respiration rate.

For BA-120 statistical study showed significant dependence of PAR and T on Chl $a$-specific $P_\mathrm{m}$ in salinities 3, 8 and 18 PSU *(ANOVA, $F_{9,32} = 2.4$, $p < 0.05$, $F_{9,32} = 3.2$, $p < 0.05$ and $F_{9,32} = 5.2$, $p < 0.001$, respectively)* and pointed to no statistically significant dependence of ecological conditions on $P_\mathrm{m}$ in salinity 13 PSU *(ANOVA, $p \geq 0.05$)*. Regarding cell-specific $P_\mathrm{m}$ there was no statistically significant influence of PAR and T on this parameter in salinity 18 PSU *(ANOVA, $p \geq 0.05$)* but was in salinity 3 PSU *(ANOVA, $F_{9,32} = 3.5$, $p < 0.05$)*, 8 PSU *(ANOVA, $F_{9,32} = 2.6$, $p < 0.05$)*, and 13 PSU *(ANOVA, $F_{9,32} = 3.0$, $p < 0.05$)*. For Chl $a$-specific $\alpha$, statistical study indicated no environmental impacts in salinities 3, 8 and 13 PSU but an impact of PAR and T in salinity 18 PSU *(ANOVA, $F_{9,32} = 2.7$, $p < 0.05$)*, while for cell-specific $\alpha$ statistical significance of PAR and T influence was obtained for all salinities *(ANOVA, $F_{9,32} = 5.1$, $p < 0.001$, ANOVA, $F_{9,32} = 2.9$, $p < 0.05$, ANOVA, $F_{9,32} = 2.5$, $p < 0.05$ and ANOVA, $F_{9,32} = 4.8$, $p < 0.001$, for salinity 3, 8, 13 and 18 PSU, respectively)*. Regarding $R_\mathrm{d}$, two-way ANOVA pointed to no environmental determination of Chl $a$-specific $R_\mathrm{d}$ values *(ANOVA, $p > 0.05$)* but it showed the influence of PAR and T on cell-specific $R_\mathrm{d}$ *(ANOVA, $F_{9,32} = 9.2$, $p < 0.001$, ANOVA, $F_{9,32} = 3.8$, $p < 0.01$, ANOVA, $F_{9,32} = 3.8$, $p < 0.01$, ANOVA, $F_{9,32} = 4.5$, $p < 0.001$, in salinities 3, 8, 13, 18 PSU, respectively)*. Tukey HSD tests pointed to some statistically significant multiple comparisons but showed a weak influence of PAR and T combined on Chl $a$-specific parameters. Regarding $\alpha$, only 3% of all multiple comparisons were statistically significant (* $p < 0.05$) with 7% of them at the highest statistical significance level (*** $p < 0.001$). For $P_\mathrm{m}$, 36% of all multiple comparisons were statistically significant (* $p < 0.05$) with 64% of them with the highest significance (*** $p < 0.001$). Regarding $R_\mathrm{d}$, as mentioned above, no statistically significant analysis of variance was indicated. Due to that, no post hoc tests were proceeded. Note that in order to shorten the text and emphasize reading, in this section the notation for the percentage of all statistically significant multiple comparisons (* p < 0.05) and the percentage of the multiple comparisons of the highest significance within the all significant comparisons (*** p < 0.001 × (* p < 0.05)–1) were written in parenthesis, one by one, separated with comma. For instance: X (15%, 20%) would mean that there were 15% of statistically significant multiple comparisons for parameter X in the post hoc tests results, whereas 20% of them were statistically the most significant. Similarly to Chl $a$-specific calculations, Tukey HSD test pointed to a selective influence of PAR and T combined on cell-specific parameters. However, this dependence was stronger when compared to Chl $a$-specific estimations ($P_\mathrm{m}$ (16%,

| 439 | 52%), $\alpha$ (19%, 43%), $R_d$ (28%, 56%)). Nonetheless, there were also some fixed relations noted for both calculation domains.

For Chl $a$-specific photosynthesis, $P_m$ increased along with PAR up to PAR of 190 µmol photons m$^{-2}$ s$^{-1}$ (Figs. S4, a, c).
Above this level $P_m$ value started to decrease slightly. This was the case in all salinities. Minimum $P_m$ was measured for cells
grown in scenario: salinity 3 PSU, T 15°C, PAR 10 µmol photons m$^{-2}$ s$^{-1}$ and it was 0.12 µmol O$_2$ (µg Chl $a$)$^{-1}$ h$^{-1}$ (Fig. S4,
a), whilst the maximum equalled 1.31 µmol O$_2$ (µg Chl $a$)$^{-1}$ h$^{-1}$ and was reached in salinity 18 PSU, T 25°C, 190 µmol
photons m$^{-2}$ s$^{-1}$ (Fig. S4, c). Dark respiration rate ($R_d$) increased with T increase and decreased with PAR increase (Figs. S5,
a, c). Minimum $R_d$ (-0.31 µmol O$_2$ (µg Chl $a$)$^{-1}$ h$^{-1}$) was measured in salinity 18 PSU, T 10°C, PAR 280 µmol photons m$^{-2}$ s$^-$
$^1$ (Fig. S5, c), while maximum (-0.02 µmol O$_2$ (µg Chl $a$)$^{-1}$ h$^{-1}$) was estimated in salinity 3 PSU, T 25°C, PAR 10 µmol
photons m$^{-2}$ s$^{-1}$ (Fig. S5, a). On the contrary, it was more difficult to determine a fixed pattern of $\alpha$ changes unequivocally.
The most fixed tendency of $\alpha$ changes was observed between all temperature-differenced scenarios in 18 PSU salinity
medium (Figs. S6, a, c). Under those conditions, it was noticeable that $\alpha$ decreased with PAR and T increase till it reached
PAR level of 190 µmol photons m$^{-2}$ s$^{-1}$. Then, $\alpha$ started to rise slowly. Regarding all gathered results (all mediums together),
minimum $\alpha$ was measured in salinity 3 PSU, T 25°C, PAR 10 µmol photons m$^{-2}$ s$^{-1}$ and equalled 0.002 µmol O$_2$ (µg Chl $a$)$^{-1}$
h$^{-1}$ [µmol photons m$^{-2}$ s$^{-1}$]$^{-1}$ (Fig. S6, a), whilst maximum was 0.013 µmol O$_2$ (µg Chl $a$)$^{-1}$ h$^{-1}$ [µmol photons m$^{-2}$ s$^{-1}$]$^{-1}$ in
salinity 13 PSU, T 10°C, PAR 10 µmol photons m$^{-2}$ s$^{-1}$. On the other hand, for cell-specific domain, $P_m$ increased along with
T and it was more pronounced in higher salinities. Concerning all results, minimum $P_m$ was 28.58 µmol O$_2$ cell 10$^{-9}$ h$^{-1}$ and,
similarly to Chl $a$-specific $P_m$ was measured in scenario: salinity 13 PSU, T 10°C, PAR 10 µmol photons m$^{-2}$ s$^{-1}$, whilst
maximum $P_m$ equalled 55.16 µmol O$_2$ cell 10$^{-9}$ h$^{-1}$ and was reached in salinity 8 PSU, T 25°C, 190 µmol photons m$^{-2}$ s$^{-1}$
(data not shown). Regarding $\alpha$, this parameter was generally negatively affected by PAR and T up to PAR of 190 µmol
photons m$^{-2}$ s$^{-1}$. However minimum value was obtained for cells growing in moderate T (salinity 8 PSU, T 20°C, PAR 10
µmol photons m$^{-2}$ s$^{-1}$) and equalled 0.81 µmol O$_2$ cell 10$^{-9}$ h$^{-1}$ [µmol photons m$^{-2}$ s$^{-1}$]$^{-1}$. Maximum $\alpha$ equalled 1.57 µmol O$_2$
cell 10$^{-9}$ h$^{-1}$ [µmol photons m$^{-2}$ s$^{-1}$]$^{-1}$ and was measured in salinity 18 PSU, T 10°C, PAR 10 µmol photons m$^{-2}$ s$^{-1}$ (Fig. S6,
d). Generally, T and PAR had a positive impact on $R_d$ for cultures grown in PAR range up to 190 µmol photons m$^{-2}$ s$^{-1}$. For
cultures grown under elevated PAR conditions, $R_d$ was lower (more intensive respiration) when compared to low PAR
scenarios. The lowest $R_d$ within all BA-120 experiments results was -16.97 µmol O$_2$ cell 10$^{-9}$ h$^{-1}$ and noted in salinity 3
PSU, T 10°C, PAR 10 µmol photons m$^{-2}$ s$^{-1}$ (Fig. S5, b), whilst the highest $R_d$ was measured in salinity 18 PSU, T 25°C,
PAR 100 µmol photons m$^{-2}$ s$^{-1}$ and equalled -2.06 µmol O$_2$ cell 10$^{-9}$ h$^{-1}$ (Fig. S5, d).
For BA-124, statistical study showed significant dependence of ecological conditions on photosynthesis parameters,
excluding Chl $a$-specific $\alpha$ *(ANOVA, p ≥ 0.05)* and cell-specific $P_m$ *(ANOVA, p ≥ 0.05)*. For the rest parameters ANOVA
results were as follows: Chl $a$-specific $P_m$ *(ANOVA, $F_{9,32}$ = 4.8, p < 0.001, ANOVA, $F_{9,32}$ = 19.7, p < 0.001, ANOVA, $F_{9,32}$ =*
*9.14, p < 0.001, ANOVA, $F_{9,32}$ = 6.5, p < 0.001 in salinity 3, 8, 13, 18 PSU, respectively)*; cell-specific $P_m$ *(ANOVA, $F_{9,32}$ =*
*7.5, p < 0.001, ANOVA, $F_{9,32}$ = 6.1, p < 0.001, ANOVA, $F_{9,32}$ = 4.3, p < 0.001 in salinity 8, 13 and 18 PSU, respectively)*;
Chl $a$-specific $\alpha$ *(ANOVA, $F_{9,32}$ = 5.0, p < 0.001, ANOVA, $F_{9,32}$ = 3.3, p < 0.01, ANOVA, $F_{9,32}$ = 3.8, p < 0.01 in salinity 3, 8*
*and 18 PSU, respectively)*; cell-specific $\alpha$ *(ANOVA, $F_{9,32}$ = 6.6, p < 0.001, ANOVA, $F_{9,32}$ = 17.9, p < 0.001, ANOVA, $F_{9,32}$ =*
*18.9, p < 0.001, ANOVA, $F_{9,32}$ = 3.1, p < 0.01, in salinity 3, 8, 13, 18 PSU, respectively)*; Chl $a$-specific $R_d$ *(ANOVA, $F_{9,32}$ =*
*10.0, p < 0.001, ANOVA, $F_{9,32}$ = 4.9, p < 0.001, ANOVA, $F_{9,32}$ = 3.8, p < 0.01, ANOVA, $F_{9,32}$ = 2.6, p < 0.05, in salinity 3, 8,*
*13, 18 PSU, respectively)*; cell-specific $R_d$ *(ANOVA, $F_{9,32}$ = 13.0, p < 0.001, ANOVA, $F_{9,32}$ = 2.2, p < 0.05, ANOVA, $F_{9,32}$ =*
*40.4, p < 0.001, ANOVA, $F_{9,32}$ = 3.1, p < 0.01)*. Post-hoc tests showed there must have been other factors, which affected the
whole process of photosynthesis as there were many not statistically significant multiple comparisons defined. Generally,
Tukey HSD tests pointed to only few statistically significant multiple comparisons, in both Chl $a$-specific, especially for $P_m$,
($P_m$ (60%, 76%), $\alpha$ (9%, 29%), $R_d$ (30%, 47%)) and cell-specific ($P_m$ (22%, 56%), $\alpha$ (34%, 63%), $R_d$ (30%, 74%)) estimations.
Nonetheless, for $P_m$ there was a tendency noted, which suggested that on average, the maximum of photosynthesis was
higher at elevated PAR. This was the case in both estimations, Chl $a$-specific and cell-specific. Maximum Chl $a$-specific $P_m$
was 3.0 and minimum 0.16 µmol O$_2$ (µg Chl $a$)$^{-1}$ h$^{-1}$. These values were measured in salinity 18 PSU in T 25°C, PAR 280
μmol photons m$^{-2}$ s$^{-1}$ and T 10°C, PAR 10 μmol photons m$^{-2}$ s$^{-1}$, respectively (Fig. S4, g). Maximum cell-specific $P_m$ was
obtained in salinity 8 PSU, T 25°C, PAR 280 μmol photons m$^{-2}$ s$^{-1}$ and minimum in salinity 13 PSU, T 20°C, PAR 10 μmol
photons m$^{-2}$ s$^{-1}$ (data not shown here). These extreme values were 53.41 and 19.17 μmol O$_2$ cell·10$^{-9}$ h$^{-1}$, respectively. It was
difficult to determine a fixed relation between ecological state and $\alpha$ changes in both domains, which was supported by the
post-hoc test (more than 91% of multiple comparisons were not statistically significant ($p \geq 0.05$) in Chl $a$-specific and more
than 35% in cell-specific estimations). Maximum Chl $a$-specific $\alpha$ was 0.02 μmol O$_2$ (μg Chl $a$)$^{-1}$ h$^{-1}$ [μmol photons m$^{-2}$ s$^{-1}$]$^{-}$
$^{1}$ and was measured in salinity 3 PSU, T 15°C, PAR 100 μmol photons m$^{-2}$ s$^{-1}$ (Fig. S6, e), while maximum cell-specific $\alpha$
(1.77 μmol O$_2$ cell 10$^{-9}$ h$^{-1}$ [μmol photons m$^{-2}$ s$^{-1}$]$^{-1}$) was obtained in salinity 13 PSU, T 10°C, PAR 10 μmol photons m$^{-2}$ s$^{-1}$.
Minimum Chl $a$-specific $\alpha$ was 0.003 μmol O$_2$ (μg Chl $a$)$^{-1}$ h$^{-1}$ [μmol photons m$^{-2}$ s$^{-1}$]$^{-1}$ and was measured in two scenarios:
salinity 3 PSU, T 10°C, PAR 280 μmol photons m$^{-2}$ s$^{-1}$ (Fig. S6, e) and salinity 18 PSU, T 15°C, PAR 10 μmol photons m$^{-2}$
s$^{-1}$ (Fig. S6, g). Minimum cell-specific $\alpha$ equalled 0.08 μmol O$_2$ cell 10$^{-9}$ h$^{-1}$ [μmol photons m$^{-2}$ s$^{-1}$]$^{-1}$ and was measured in
salinity 18 PSU, T 15°C, PAR 190 μmol photons m$^{-2}$ s$^{-1}$ (Fig. S6, h). Similarly to $\alpha$, it was difficult to determine fixed
relations between PAR and T and $R_d$, which was supported by statistics (about 70% of multiple comparisons for both Chl $a$-
specific and cell-specific $R_d$ were not statistically significant (Tukey HSD, $p \geq 0.05$)). Nonetheless, it was observed that,
generally, $R_d$ decreased along with PAR increase in cell-specific estimations. Maximum Chl $a$-specific and cell-specific $R_d$
was -0.03 μmol O$_2$ (μg Chl $a$)$^{-1}$ h$^{-1}$ and -1.52 μmol O$_2$ cell 10$^{-9}$, respectively. These values were obtained in salinity 13 PSU,
T 20°C, PAR 10 μmol photons m$^{-2}$ s$^{-1}$ and salinity 18 PSU, T 20°C, PAR 190 μmol photons m$^{-2}$ s$^{-1}$, respectively for Chl $a$-
and cell-specific calculations. Minimum Chl $a$-specific $R_d$ was measured in salinity 13 PSU, T 10°C, PAR 280 μmol photons
m$^{-2}$ s$^{-1}$ and was -0.27 μmol O$_2$ (μg Chl $a$)$^{-1}$ h$^{-1}$, whilst minimum cell-specific $R_d$ was measured in salinity 13 PSU, T 10°C,
PAR 10 μmol photons m$^{-2}$ s$^{-1}$ and equalled -12.19 μmol O$_2$ cell 10$^{-9}$ h$^{-1}$ (data not shown here).
For BA-132, statistical study showed significant dependence of PAR and T on Chl $a$- and cell-specific $P_m$ *(for Chl a-*
*specific: ANOVA, $F_{9,32}$ = 6.2, p < 0.001, ANOVA, $F_{9,32}$ = 23.1, p < 0.001, ANOVA, $F_{9,32}$ = 25.2, p < 0.001, ANOVA, $F_{9,32}$ =*
*16.0, p < 0.001; for cell-specific: ANOVA, $F_{9,32}$ = 4.8, p < 0.001, ANOVA, $F_{9,32}$ = 24.3, p < 0.001, ANOVA, $F_{9,32}$ = 24.3, p <*
*0.001, ANOVA, $F_{9,32}$ = 21.2, p < 0.001; all numbers given for salinities 3, 8, 13, 18 PSU, respectively)*. Regarding other Chl
$a$-specific parameters, there were no statistically significant impacts of PAR and T on $\alpha$ in salinities 3, 13, 18 PSU *(ANOVA,*
*p ≥ 0.05)* but were in salinity 8 PSU *(ANOVA, $F_{9,32}$ = 2.7, p < 0.05)* and no impacts on Chl $a$-specific $R_d$ in salinities 3, 8, 18
PSU *(ANOVA, p ≥ 0.05)* but were in salinity 13 PSU *(ANOVA, $F_{9,32}$ = 2.8, p < 0.05)*. Regarding other than $P_m$ cell-specific
parameters, there was no ecological determination of $\alpha$ noted in salinities 3 and 8 PSU and of $R_d$ in salinity 13 PSU *(ANOVA,*
*p ≥ 0.05)*, while there were statistically significant environmental impacts calculated for $\alpha$ in salinity 13 PSU *(ANOVA, $F_{9,32}$*
*= 3.2, p < 0.01)* and 18 PSU *(ANOVA, $F_{9,32}$ = 2.9, p < 0.05)* and for $R_d$ in salinities 3, 8 and 18 PSU *(ANOVA, $F_{9,32}$ = 3.2, p*
*< 0.05, ANOVA, $F_{9,32}$ = 3.1, p < 0.01, ANOVA, $F_{9,32}$ = 2.4, p < 0.05, respectively)*. Tukey HSD tests pointed to statistically
significant multiple comparisons, in both Chl $a$-specific and cell-specific maximum of photosynthesis ($P_m$ (68%, 85%), $P_m$
(62%, 76%), respectively). Post hoc tests indicated no significant multiple comparisons for Chl $a$-specific $\alpha$ (>1%, >1%), a
few significant multiple comparisons for Chl $a$-specific $R_d$ (8%, 38%), cell-specific $\alpha$ (18%, 67%) and cell-specific $R_d$ (6%,
20%)). It was observed, that in cell-specific estimations, $P_m$ increased along with PAR increase, while $\alpha$ decreased at
elevated PAR. It was the most difficult to determine a fixed tendency for the $R_d$ response to changing environmental
conditions. This was supported by statistical tests (Tukey HSD, more than 93% of multiple comparisons were not
statistically significant ($p \geq 0.05$)). Maximum cell-specific $P_m$ was 158.94 μmol O$_2$ cell 10$^{-9}$ h$^{-1}$ and was reached in salinity 8
PSU, T 25°C, PAR 280 μmol photons m$^{-2}$ s$^{-1}$, whilst minimum equalled 28.04 μmol O$_2$ cell 10$^{-9}$ h$^{-1}$ in salinity 18 PSU, T
15°C, PAR 10 μmol photons m$^{-2}$ s$^{-1}$ (Fig. S4, l). Maximum cell-specific $\alpha$ was 1.78 μmol O$_2$ cell 10$^{-9}$ h$^{-1}$ [μmol photons m$^{-2}$
s$^{-1}$]$^{-1}$ and was measured in salinity 13 PSU, T 20°C, PAR 10 μmol photons m$^{-2}$ s$^{-1}$, while minimum was reached in salinity
18 PSU, T 20°C, PAR 100 μmol photons m$^{-2}$ s$^{-1}$ and equalled 0.19 μmol O$_2$ cell 10$^{-9}$ h$^{-1}$ [μmol photons m$^{-2}$ s$^{-1}$]$^{-1}$ (Fig. S6, l).
Regarding cell-specific $R_d$, maximum was measured in salinity 18 PSU, T 15°C, PAR 100 μmol photons m$^{-2}$ s$^{-1}$ and equalled
-3.17 μmol O$_2$ cell 10$^{-9}$ h$^{-1}$ (Fig. S5, l), whilst minimum was -15.55 μmol O$_2$ cell 10$^{-9}$ h$^{-1}$ and was obtained in salinity 3
PSU, T 10°C, PAR 10 µmol photons $m^{-2} s^{-1}$ (Fig. S5, j). For Chl $a$-specific $P_m$, the increases along with T and salinity was
observed, whilst $\alpha$ presented strong changing characteristics between scenarios. The fixed influence of PAR and T on $\alpha$
values was difficult to determine, which was supported by statistics *(ANOVA, $p \geq 0.05$)*. Contrary to the above, it was plainly
evident that PAR increase had a negative impact on Chl $a$-specific $R_d$. Maximum Chl $a$-specific $P_m$ was 6.22 µmol $O_2$ (µg
Chl $a$)$^{-1}$ h$^{-1}$ and was reached in salinity 18 PSU, T 25°C, PAR 280 µmol photons $m^{-2} s^{-1}$ (Fig. S4, k), whilst minimum
equalled 0.12 µmol $O_2$ (µg Chl $a$)$^{-1}$ h$^{-1}$ in salinity 3 PSU, T 25°C, PAR 10 µmol photons $m^{-2} s^{-1}$ (Fig. S4, i). Maximum Chl
$a$-specific $\alpha$ was 0.02 µmol $O_2$ (µg Chl $a$)$^{-1}$ h$^{-1}$ [µmol photons $m^{-2} s^{-1}$]$^{-1}$ and was measured in salinity 18 PSU, T 15°C, PAR
10 µmol photons $m^{-2} s^{-1}$ (Fig. S6, k), while minimum was reached in salinity 3 PSU, T 15°C, PAR 10 µmol photons $m^{-2} s^{-1}$
and equalled 0.003 µmol $O_2$ (µg Chl $a$)$^{-1}$ h$^{-1}$ [µmol photons $m^{-2} s^{-1}$]$^{-1}$ (Fig. S6, i). Concerning Chl $a$-specific $R_d$, maximum
was measured in salinity 3 PSU, T 20°C, PAR 10 µmol photons $m^{-2} s^{-1}$ and equalled -0.02 µmol $O_2$ cell $10^{-9}$ h$^{-1}$ (Fig. S5, i),
whilst minimum was -0.39 µmol $O_2$ cell $10^{-9}$ h$^{-1}$ and was obtained in salinity 13 PSU, T 25°C, PAR 280 µmol photons $m^{-2} s^{-1}$
. Generally, in both domains, photosynthesis parameters were the highest for BA-132 when compared to other strains.
The analysis of photosynthesis characteristics enabled examining and defining the photoacclimation process of all three
strains of *Synechococcus* sp. This was done on the basis of the photosynthetic parameters (Figs. S4-S6) and Photosynthesis-
Irradiance (*P-E*) curves (exemplification shown in Fig. 6). The curves were plotted on the basis of laboratory results (Clark
oxygen electrode measurements) using the equation of Jassby and Platt (1976). According to a photoacclimation model
description (Prezelin, 1981; Prezelin and Sweeney, 1979; Ramus, 1981; Richardson et al., 1983; Pniewski et al., 2016), the
results of the present study indicated changes in Photosynthetic Units (PSU) sizes as the photoacclimation mechanism,
which occurred most frequently (Table 1). There were also *P-E* curves pointing to some changes in enzymatic reactions and
the altering of accessory pigments activity. Changes in PSU numbers were noted as well, but these observations were
episodic. In this paper the term 'OTHER' stands for changes in enzymatic reactions and the altering of accessory pigments
activity and concerns photoacclimation mechanisms other than changes in PSU sizes (PSUsize) or changes in PSU number
(PSUno.). In general, photoacclimation did not occur in low-saline medium (salinity 3). According to the results,
photoacclimation mechanisms were observed in only four scenarios with low salinity: BA-120 25°C salinity 3 PSU, BA-124
25°C salinity 3 PSU, BA-132 10°C salinity 3 PSU, and BA-132 25°C salinity 3 PSU. For BA-120, photoacclimation
occurred more frequently at higher T (20 and 25°C) than lower T (10 and 15°C). However, if it had been observed in low T
conditions, it usually stood for OTHER, not for PSUsize or PSUno. For BA-124 and BA-132 photoacclimation was noted in
the whole T range. All photoacclimation mechanisms observed for different strains are listed in Table 1.

**4 Discussion**

Picoplanktonic organisms show a lot of adaptations, which enable them to spread in aquatic environments (e.g., Stomp et al.,
2007; Jodłowska and Śliwińska, 2014; Larsson et al., 2014; Jasser and Callieri, 2017). What is more, picocyanobacteria
often dominate and occupy the niches, which are inaccessible for other photoautotrophs. Owing to the fact that PCY are
small-sized cells and consequently possess an advantageous surface area to volume ratio, they can assimilate trace amount of
nutrients and effectively absorb light. Therefore, in oligotrophic regions of seas and oceans PCY compete with other
cyanobacteria and microalgae and it can determine primary production of the whole marine ecosystem (Six et al., 2007a;
Richardson and Jackson, 2007; Worden and Wilken, 2016). This is also true for eutrophic basins (Stal et al., 2003;
Haverkamp et al., 2008; 2009; Callieri, 2010; Mazur-Marzec et al., 2013).
The distribution of PCY are determined by their optimal ecological requirements for light and temperature. Due to the
presented results, PAR and T had positive effects on the number of cells for two out of the three studied strains of
*Synechococcus* sp. The highest cell concentrations were noted in scenarios with the highest T (25°C) and the highest PAR
level (280 µmol photons $m^{-2} s^{-1}$) for BA-124 and BA-132. The BA-120 strain behaved differently when compared to the
other strains. For BA-120, the decrease in number of cells was observed in high PAR conditions, i.e. cell abundances for red
strain cultures grown under the most elevated PAR were lower than the number of BA-120 cells measured in cultures grown
under 190 µmol photons $m^{-2} s^{-1}$. According to the results derived from pigmentation, Chl $a$ fluorescence and photosynthesis
sections of the present study, the decrease in number of cells under the elevated PAR could have likely been associated with
Photosystem II photo-inhibition. This is a conclusion of a few observations, which are as follows. Firstly, there was a higher
cell-specific Car content observed for 280 µmol photons $m^{-2} s^{-1}$ when compared to 190 µmol photons $m^{-2} s^{-1}$. Secondly,
higher $F_v/F_m$ values were observed for 280 µmol photons $m^{-2} s^{-1}$ when compared to 190 µmol photons $m^{-2} s^{-1}$, especially for
low T scenarios and for all scenarios in the lowest salinity medium. Thirdly, for Chl $a$-specific photosynthesis, $P_m$ increased
along with PAR until 190 µmol photons $m^{-2} s^{-1}$, above which the values started to decrease slightly in all salinity mediums.
According to the above, a PAR level of 190 µmol photons $m^{-2} s^{-1}$ could be defined as the PSII photo-inhibition point for the
red strain. This implies BA-120 did not lead as effective photosynthesis being grown in PAR of more than 190 µmol photons
$m^{-2} s^{-1}$ as the cells grown in PAR levels equal or are beneath 190 µmol photons $m^{-2} s^{-1}$.
Cyanobacteria are generally recognized to prefer low light intensity for growth (Fogg and Thake, 1987; Ibelings, 1996).
Some picoplanktonic organisms demonstrated the ability to survive and resume growth after periods of total darkness. Such
a pronounced capacity for survival in the dark would enable these organisms to outlive the seasonal rhythm of winter
darkness and sinking into the aphotic zone (Antia, 1976). The investigated strains of *Synechococcus* sp. were found to be
well adapted to relatively low and high PAR levels. The latter was especially evident at the high treatment T. This
conclusion is consistent with the observations of picocyanobacteria maximum abundance at the euphotic zone in coastal and
offshore marine waters (Stal et al., 2003; Callieri, 2010). Moreover, Kana and Glibert (1987a,b) showed that *Synechococcus*
sp. could grow at irradiance as high as 2000 µmol photon $m^{-2} s^{-1}$. Regarding the comparison of abundance values of the
analyzed strains, the results showed that in all synthetically developed environmental scenarios, BA-124 was the strain of the
highest cell abundance. This is consistent with the Baltic Sea field studies (Mazur-Marzec et al., 2013).
Surface and near-surface populations experience extremely variable light and temperature conditions (Millie et al.,
1990), and these factors are the ones that affect the composition of photosynthetic pigments and photosynthesis performance
of PCY (Jodłowska and Śliwińska, 2014). Picocyanobacteria with a high concentration of PC are chromatically better
adapted to harvest longer wavelengths of PAR than those with PE as a dominating pigment. Therefore, such PCY, such as
the BA-124 strain, usually dominate in surface euphotic waters (Stal et al., 2003; Haverkamp et al., 2008; 2009). On the
other hand, the strains rich in PE (BA-120 and BA-132), usually occurred deeper (Fahnenstiel et al., 1991; Hauschild et al.,
1991; Vörös et al., 1991). Nonetheless, generally PCY, thanks to their high concentration of photosynthetic pigments, may
occur in waters under low light intensity (Stal et al., 2003). Carotenoids have a dual role in the cell: to maintain a high
capacity for photosynthetic light absorption and to provide protection against photooxidation (Siefermann-Harms, 1987).
This feature additionally explains why picoplanktonic *Synechococcus* is able to grow successfully both in the surface layer
of the sea and also in deeper waters (Stal and Walsby, 2000; Stal et al., 2003). This research showed that regarding BA-120
cell-specific pigments content, there were very high concentrations of Chl $a$ observed in the whole T range under low PAR.
This could have implied the photoacclimation type, which was the change in PSU number. This mechanism was observed in
$P$-$E$ curves for scenario with salinity 8 PSU and temperature 20°C.
PAR and T were the main factors also in terms of influencing the changes in Chl $a$ fluorescence in three strains of
*Synechococcus* sp. This may likely be linked to a great importance of PCY domination in many aquatic ecosystems during
the summer period. Due to Chl $a$ fluorescence parameters results, it should be noted that PAR increase always had a negative
impact on ΦPSII, which implied that cells, previously acclimated to high light conditions, had lower PSII photosystem
efficiency under actinic light.
The results showed that T, PAR and salinity influenced the photosynthesis parameters only to a certain degree. There
were many not statistically significant multiple comparisons pointed by post hoc tests. However, it was found that generally,
in cell-specific estimations, elevated PAR had a negative effect on $\alpha$ and PAR increase and influenced the respiration
negatively. For each of the studied strains of *Synechococcus* sp., the highest $\alpha$ and the lowest $R_d$ were noted for the cells
grown under the lowest PAR (10 μmol photons m$^{-2}$ s$^{-1}$). On the other hand, the highest values of $P_m$ were noted at the
highest PAR. It pointed to inability for the cells incubated in low PAR conditions to be as effective in photosynthesis as the
cells grown under high irradiances. On the basis of P-E curves derived in this study, three types of photoacclimation
mechanisms of *Synechoccocus* sp. were observed: change in PSU size, change in PSU number and altering accessory
pigments activity and changes in enzymatic reactions. This was a striking observation because in the literature the two first
of photoacclimation mechanisms listed above are predominant (Stal et al., 2003; Jodłowska and Śliwińska, 2014). The
present study showed that changes in PSU size occur most frequently (Table 1). The second, ranked by frequency of
occurrence, was the altering of accessory pigment activity. PSU number changes in *Synechoccocus* sp. occurred rarely,
which is consistent with literature (Jodłowska and Śliwińska, 2014). Moreover, in this study, salinity 3 PSU was the
medium, where the photoacclimation mechanisms in the *Synechococcus* sp. cells were recognized the least frequently. The
changes of photosynthesis parameters ($P_m$, $\alpha$, $R_d$) under different environmental conditions explains the occurrence of
different photoacclimation mechanisms. According to the results, *Synechococcus* strains present different ecophysiological
characteristics, however, they all demonstrate the tolerance to elevated PAR (for BA-120 to a certain degree) and T levels
and could have effectively acclimated to varied water conditions. These strains were able to change the composition of
photosynthetic pigments in order to use light quanta better. The ability of *Synechococcus* sp. to sustain its growth in low light
conditions and its low photoinhibition in exposure to high light intensities could give PCY an advantage over the other
phytoplankton in optically changing waters (Jasser, 2006).

Due to occurrence of extremes in salinity and other environmental conditions in the Baltic Sea area, the Baltic

inhabitants are highly adapted to different regions and often reach their physiological limits (Snoeijs-Leijonmalm and
Andrén, 2017). The changing environmental conditions the cultures were grown in during the experiments were salinity, T
and PAR. Daily mean sea surface temperature (Leppäranta and Myrberg, 2009) presents strongly pronounced annual cycles
in the Baltic Sea area. Sea surface temperature (SST) range between about 10 and 20ºC may be timed in the Baltic between
June and September with some inter-annual changes (Siegel and Gerth, 2017). SSTs reaching and exceeding 20ºC are also
observed in the Baltic basin. For instance, according to Siegel and Gerth (2017), SSTs higher than 20ºC were recorded in
almost whole Baltic area beyond Danish Straits, Bothnian Bay and northern Bothnian Basin in the warmest week of 2016, in
July. According to above, the temperatures, under which the picocyanobacterium cultures were grown in the present study
(10 – 25ºC) can be defined as representative for the Baltic Sea. Furthermore, the salinity ranges applied in the experiment are
also Baltic's representatives. The Baltic Sea horizontal salinity gradient is high and different sub-basins are characterized by
different mean salinity values. The gradient decreases North towards. The highest salinity is observed in the Baltic Sea
boundary to the North Sea (Skagerrak, mean salinity ranges between 28.34 and 32.71), while the lowest mean salinity is
observed in the Baltic northernmost regions (around 2.35 – 3.96 in Bothnian Basin). These numbers were determined on the
basis of climatological data from the Baltic Atlas of Long-Term Inventory and Climatology (Feistel et al., 2008; 2010).
Thus, the presented analysis may derive accurate assumptions regarding the regional distribution of *Synechococcus* sp.
strains in the Baltic Sea. For instance, a salinity horizontal gradient can be one of the factors determining the abundance of a
certain strain in the basin. More saline waters are most preferred by BA-132. On that basis, one can assume the concentration
of this strain will be higher near the Baltic Sea entrance (Danish Straits) than in Bothnian Bay. Additionally, it was observed
that despite elevated PAR conditions being more suitable for BA-124 and BA-132 to grow intensively, all analyzed strains
were able to survive and grow in low PAR conditions. This is consistent with other previously published Baltic studies (Stal
et al., 2003; Jodłowska and Śliwińska, 2014) stating that this is caused by phycobilisomes, which are structural components
of picocyanobacteria PSII photosystem. The presence of PCY cells throughout the whole euphotic water column was also
reported in limnological studies (Becker et al., 2004, Callieri, 2007).

The discrepancies between the strains ecophysiology derived in this study amplified the need for in-depth investigation

of three strains separately. What is more, according to the author's best knowledge, Baltic brown strain (BA-132) is the least
recognized strain out of three analyzed *Synechococcus* sp. strains, so far. Stal et al. (2003) and Haverkamp et al. (2008)

pointed to its inhabitation in the Baltic Sea but did not give its characteristics in detail. In the recent research more detailed investigation on BA-132 was provided (Jodłowska and Śliwińska, 2014). Nonetheless, the autecology issue of this strain still requires careful studies. The present paper derives the new knowledge on the BA-132 responses to changing ecological conditions. What is more, the study places BA-132 among the other *Synechococcus* sp. strains and compares their ecophysiology pointing to significant differences between these organisms.

The study of Baltic picoplankton ecophysiology is also of a great importance in the context of climate change. According to Belkin (2009), the Baltic Sea is among the Large Marine Ecosystems (LME), where the most rapid warming is being observed (the increase in SST between 1982 and 2006 > 0.9°C). Moreover, there are studies pointing to an increase of average winter temperatures in northern Europe by several degrees by the year 2100 (Meier, 2002). These along with the presented results, which suggest that all analyzed strains of *Synechococcus* sp. were positively affected by T can be a strong argument for further numerical research on examining the effect of long-term positive temperature trend on the abundance of PCY in the Baltic Sea (the need for picoplankton model representation). What is more, the feedback relation, which is the surface most layer being warmed up by irradiance trapped in the cells of phytoplankton may derive interesting conclusions on the functioning of the ecosystem and the living organisms being the internal source of heat in the marine medium.

The observation that T increase had a positive impact on all strains' number of cells is also consistent with field studies, which indicate the seasonal cycle of PCY maximal abundances (Flombaum et al., 2013; Dutkiewicz et al., 2015; Worden and Wilken, 2016). Hajdu et al. (2007) showed that during the decline phase of Baltic cyanobacterial blooms in late summer, unicellular and colony-forming picocyanobacteria increased in abundance. Mazur-Marzec et al. (2013) indicated that the contribution of PCY biomass in total summer cyanobacterial biomass was usually high and ranged from 20% at the beginning of July to 97% in late July and August. Moreover, Paczkowska et al. (2017) pointed to the abundance of 40-90% in the summertime in the Baltic Sea and to PCY being a dominant size group in all Baltic basins. Stal et al. (1999) reported that 65% of the phytoplankton-associated Chl *a* concentration in the Baltic Proper during late summer belonged to picoplankton, while the second most dominant group was nitrogen-fixing cyanobacteria (*Aphanizomenon* sp., *Dolichospermum* sp. and *Nodularia* sp.). Contrary to that, there were also some reports regarding high PCY abundance in the wintertime. For instance, during the winter–spring period, PCY was the second most dominant fraction in the Baltic Sea (Paczkowska et al., 2017). The present study showed that PCY can survive and grow also in low T and PAR conditions, which is consistent to the finding of Paczkowska et al. (2017).

The studies of autecology of the PCY community and an understanding of its response to main environmental factors is an important step in recognizing the phenomenon of PCY blooms in marine environments. Additionally, the laboratory experiments became a foundation in developing a new approach to Baltic Sea phytoplankton modeling - development of pico-bioalgorithm describing PCY growth, which may enable long-term numerical studies on the response of PCY to changing environment.

**5 Conclusions**

Discrepancies in number of cells, pigmentation changes, Chl *a* fluorescence and photosynthesis characteristics implied that BA-120, BA-124 and BA-132 should be studied and examined separately.

Nonetheless, there were also fixed features referring to all analyzed strains, reasoning the association these features with *Synechococcus* as a species, in general. For instance, according to the derived results, PAR and T played a key role in the life cycle of all three strains. Additionally, the positive impact of salinity on the number of cells was observed in each culture. Another similarity was the prevalence of one of photoacclimation mechanisms, which was the change in size of PSU. This second most frequent type was altering of accessory pigments and the least frequent was the change in PSU number.

Contrary to that, the main differences were: different responses of number of cells to respective environmental conditions in different cultures; various photoacclimation mechanisms observed; and different changes in pigmentation.

According to the latest research, PCY are a great contributor to total primary production in the Baltic Sea and may contribute to summer cyanobacteria bloom at a high degree. This explains the authors' motivation to lead an in-depth investigation on Baltic PCY response to a changing environment. The present research is a first step on the way to deriving new knowledge on *Synechococcus* sp. ecophysiology and is a foundation for further studies.

**Acknowledgments**

The authors would like to thank the Reviewers and Editor for their valuable comments and suggestions to improve the quality of the paper. The authors would like to thank Simon Bretherton for English language support and Proof Reading Service company for professionally proofread. The authors gratefully thank Jakub Maculewicz (IO UG), for his excellent and professional technical assistance. The author SSW was financially supported by BMN grants, Poland, no. 538-G245-B568-17. This work has been funded by the Polish National Science Centre project (contract number: 2012/07/N/ST10/03485) entitled: "Improved understanding of phytoplankton blooms in the Baltic Sea based on numerical models and existing data sets". The author (AC) received funding from Polish National Science Centre in a doctoral scholarship program (contract number: 2016/20/T/ST10/00214). AC contribution was also supported by the statutory funding of IO PAS.

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

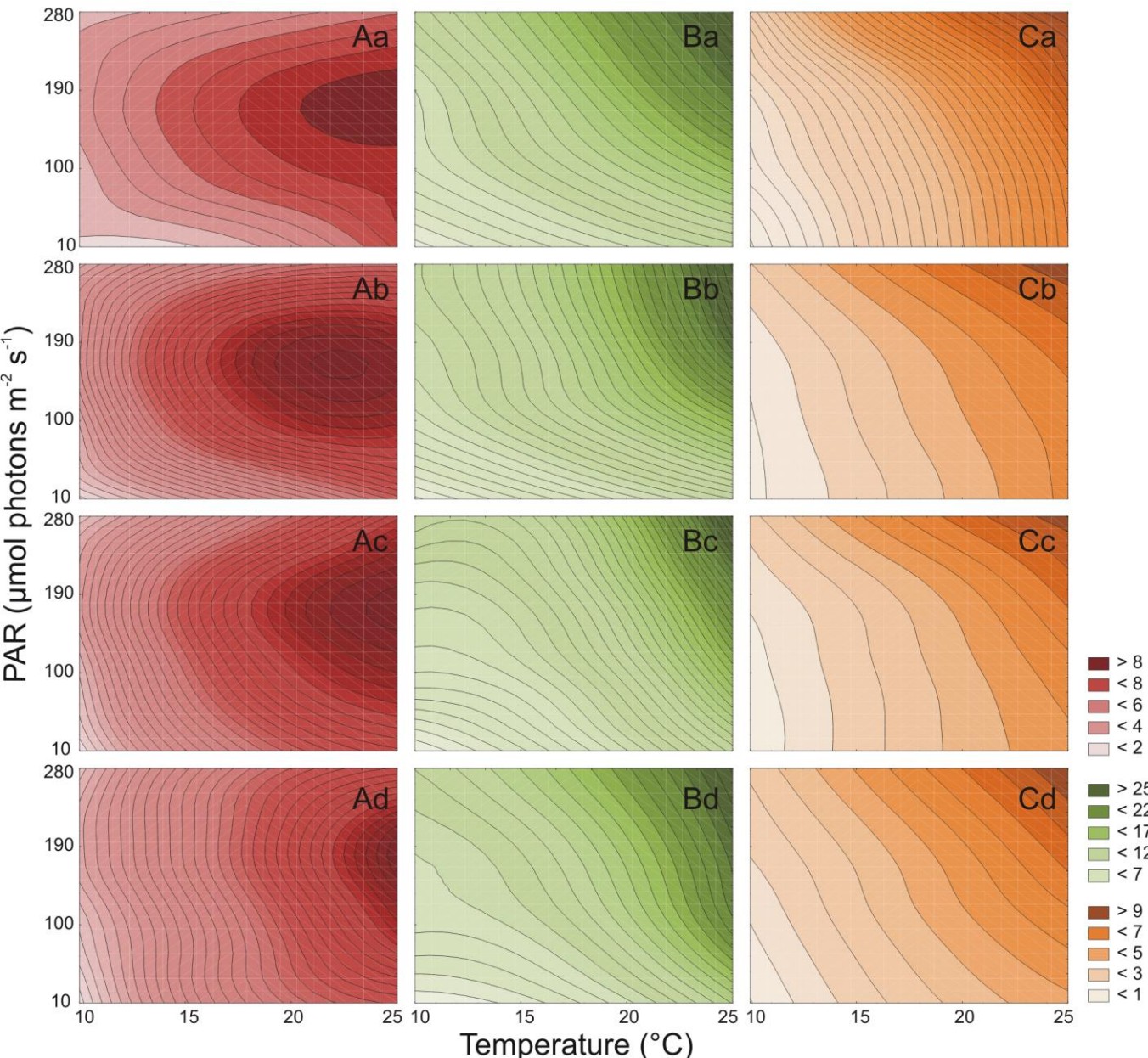


**Figure 1.** Cell number ($10^6$ cell $mL^{-1}$) for three *Synechococcus* sp. strains: BA-120 (A), BA-124 (B) and BA-132 (C) under
different PAR and temperature conditions in 4 salinity mediums: 3 PSU (a), 8 PSU (b), 13 PSU (c) and 18 PSU (d).


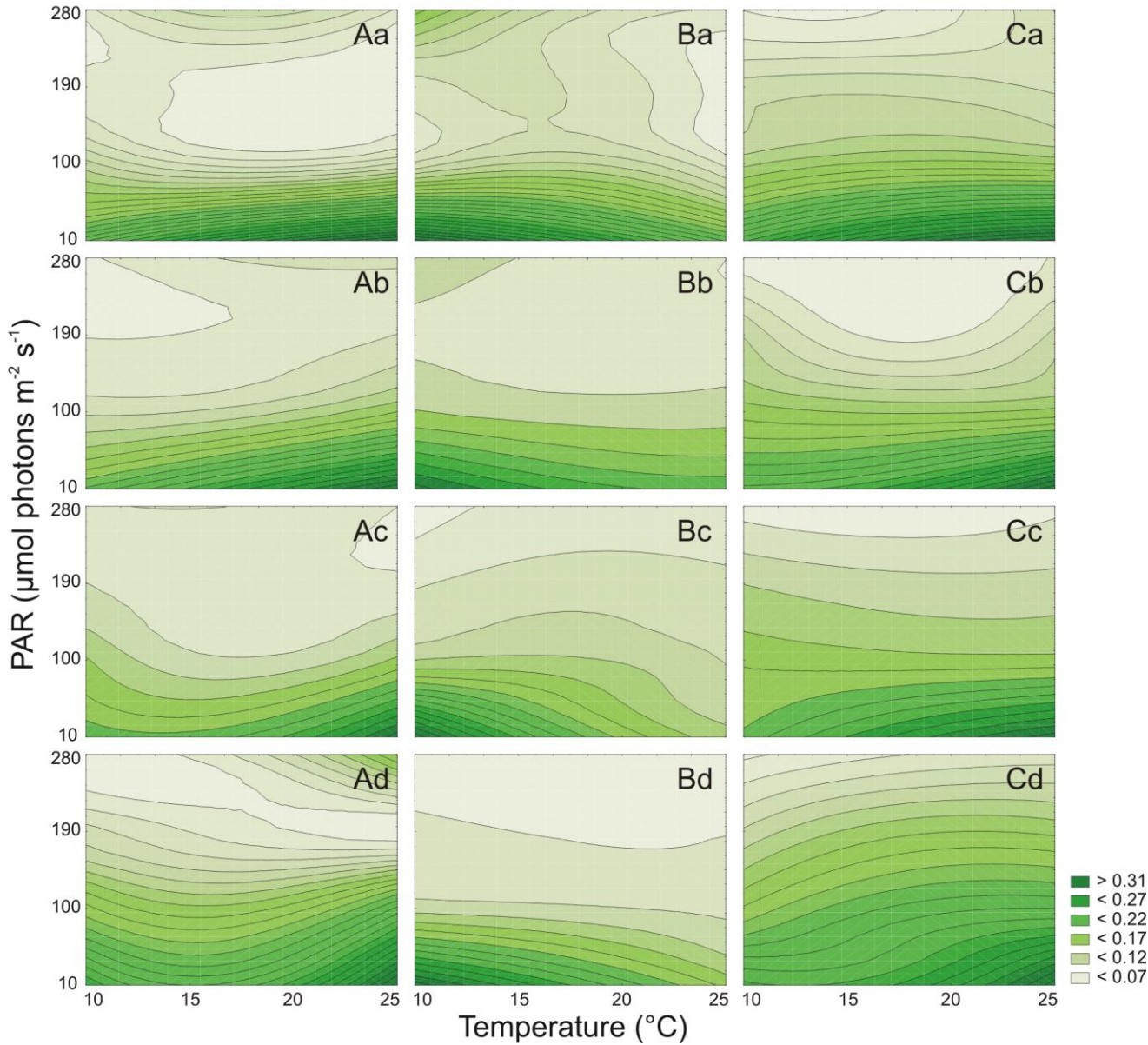


**Figure 2.** Cell-specific Chl *a* (pg cell$^{-1}$) changes for three *Synechococcus* sp. strains: BA-120 (A), BA-124 (B) and BA-132
(C) under different PAR and temperature conditions in 4 salinity mediums : 3 PSU (a), 8 PSU (b), 13 PSU (c) and 18 PSU
(d).

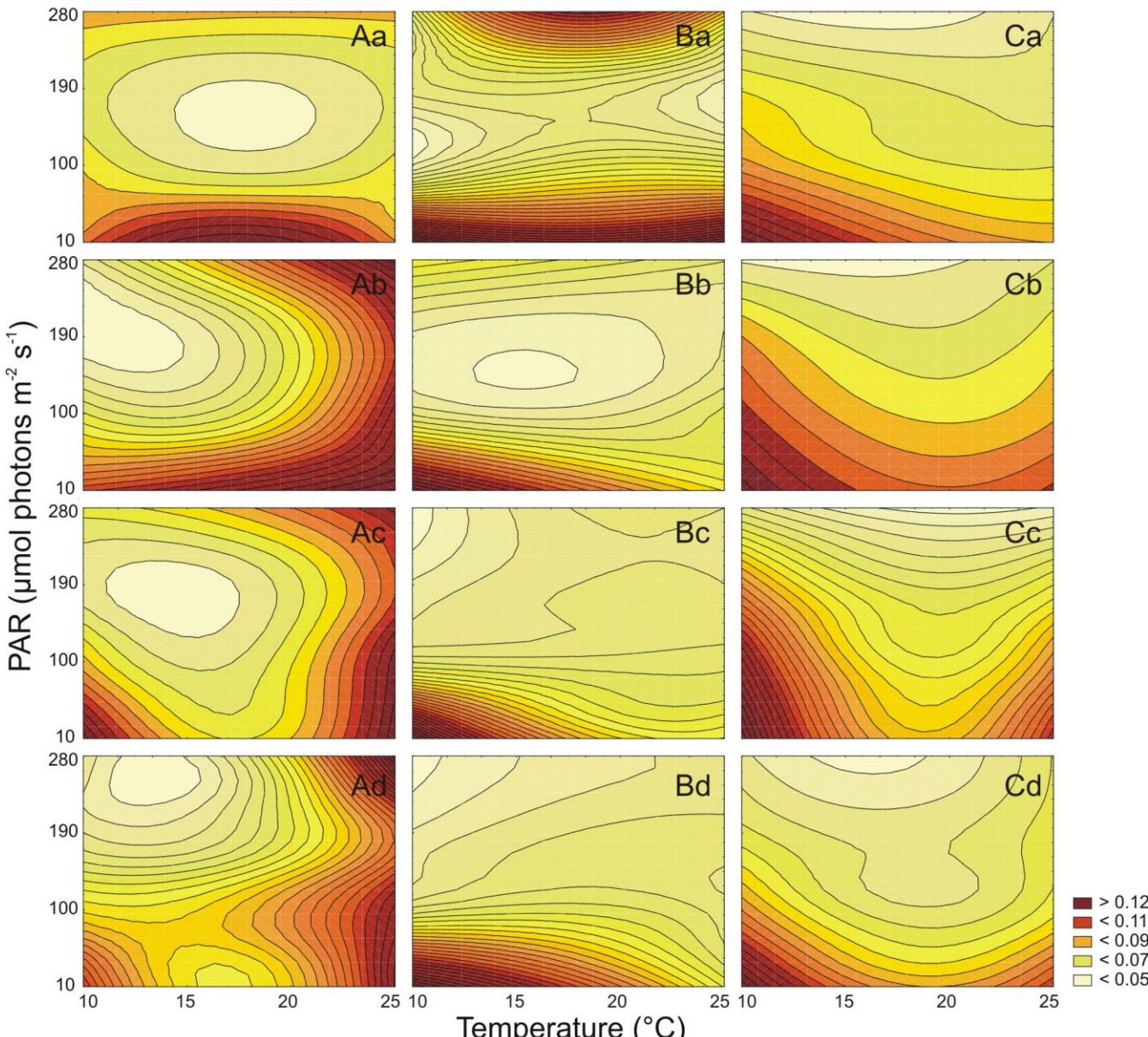

**Figure 3.** Cell-specific Car (pg cell$^{-1}$) changes for three *Synechococcus* sp. strains: BA-120 (A), BA-124 (B) and BA-132
(C) under different PAR and temperature conditions in 4 salinity mediums: 3 PSU (a), 8 PSU (b), 13 PSU (c) and 18 PSU
(d).

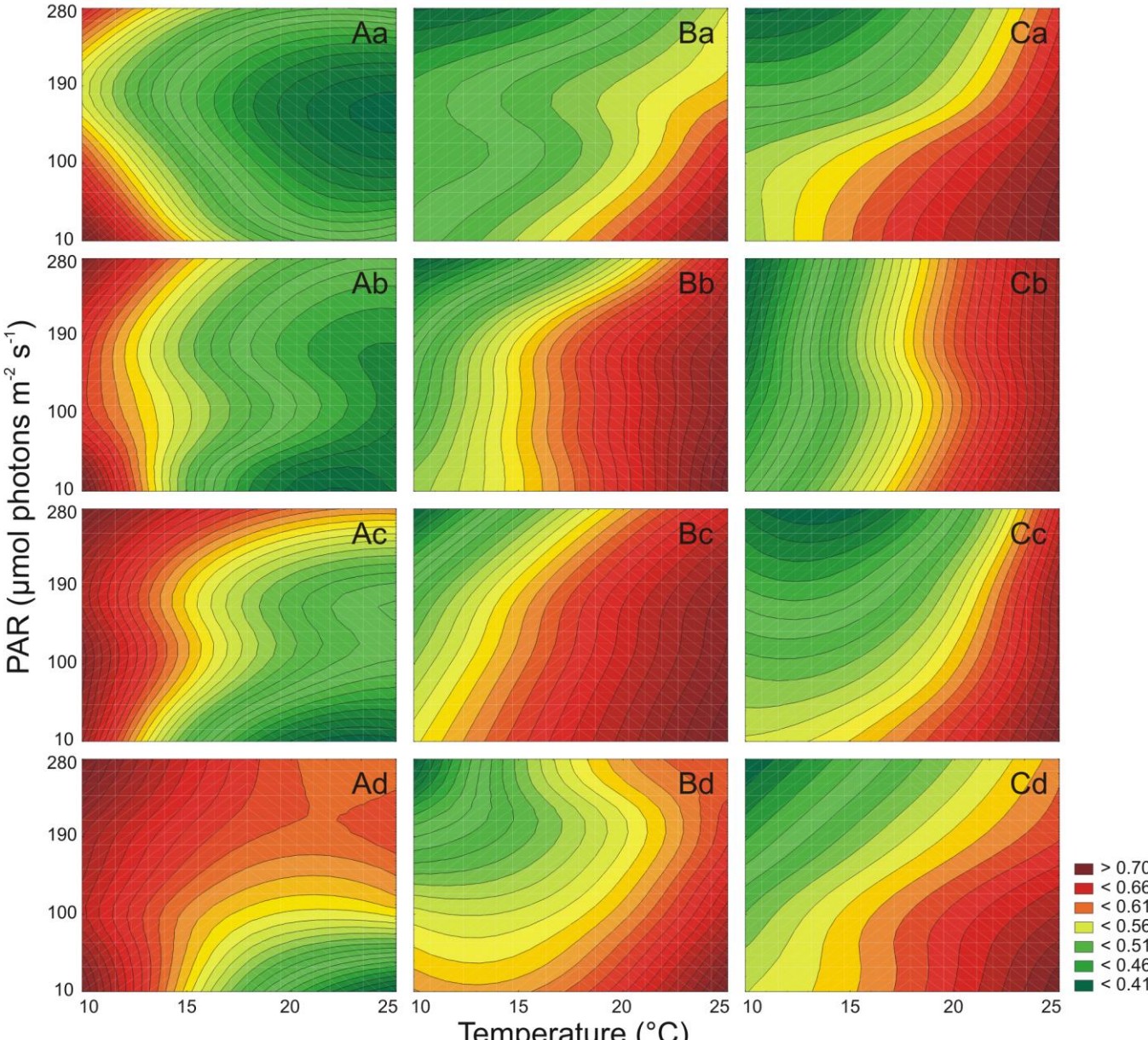

927

**Figure 4.** The maximum photochemical efficiency of PSII in the dark-adapted state ($F_v/F_m$) for three *Synechococcus* sp.
strains: BA-120 (A), BA-124 (B) and BA-132 (C) under different PAR and temperature conditions in 4 salinity mediums: 3
PSU (a), 8 PSU (b), 13 PSU (c) and 18 PSU (d).

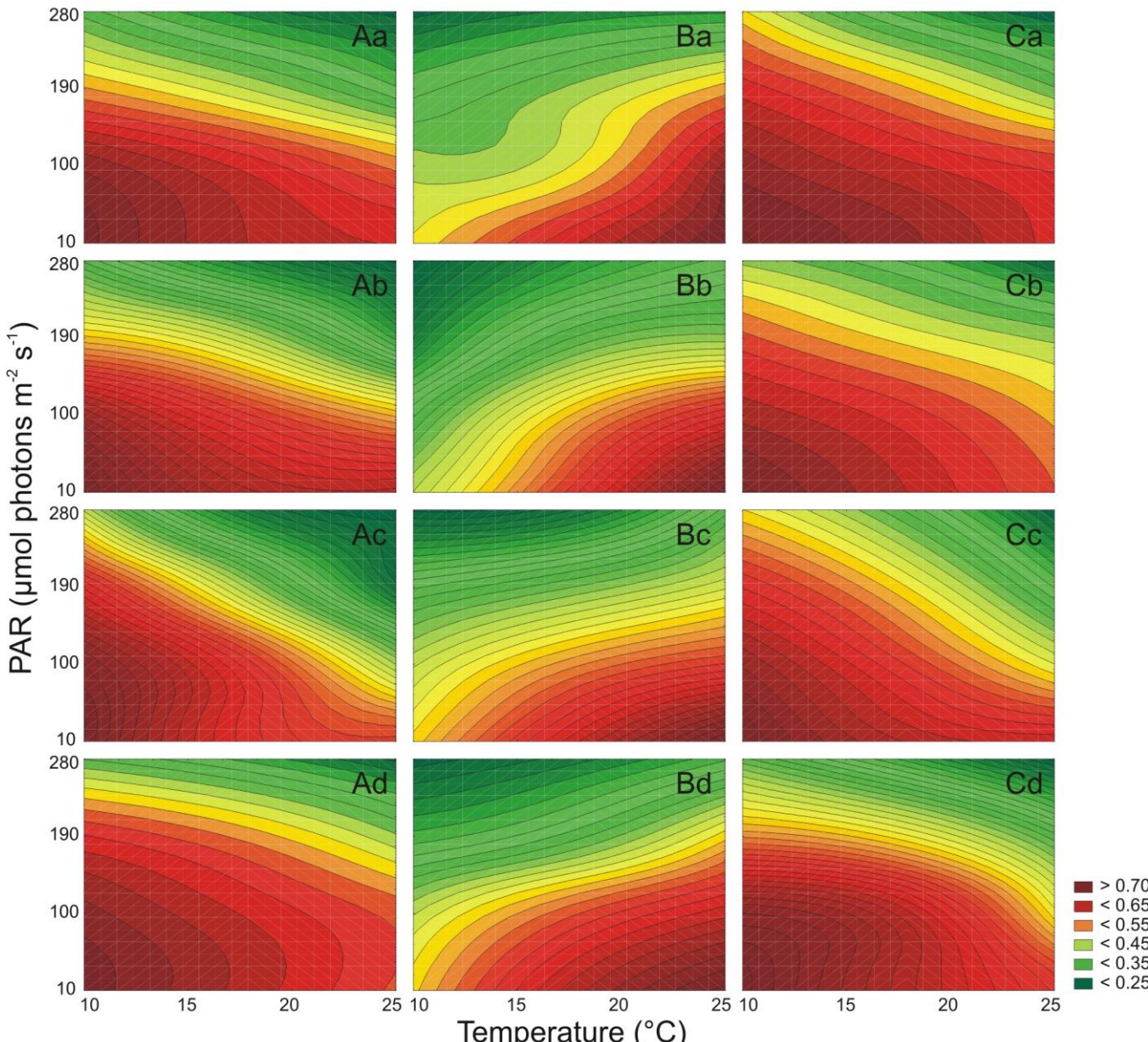

**Figure 5.** The photochemical efficiency of PSII under actinic light intensity (ΦPSII) for three *Synechococcus* sp. strains: BA-120 (A), BA-124 (B) and BA-132 (C) under different PAR and temperature conditions in 4 salinity mediums: 3 PSU (a), 8 PSU (b), 13 PSU (c) and 18 PSU (d).

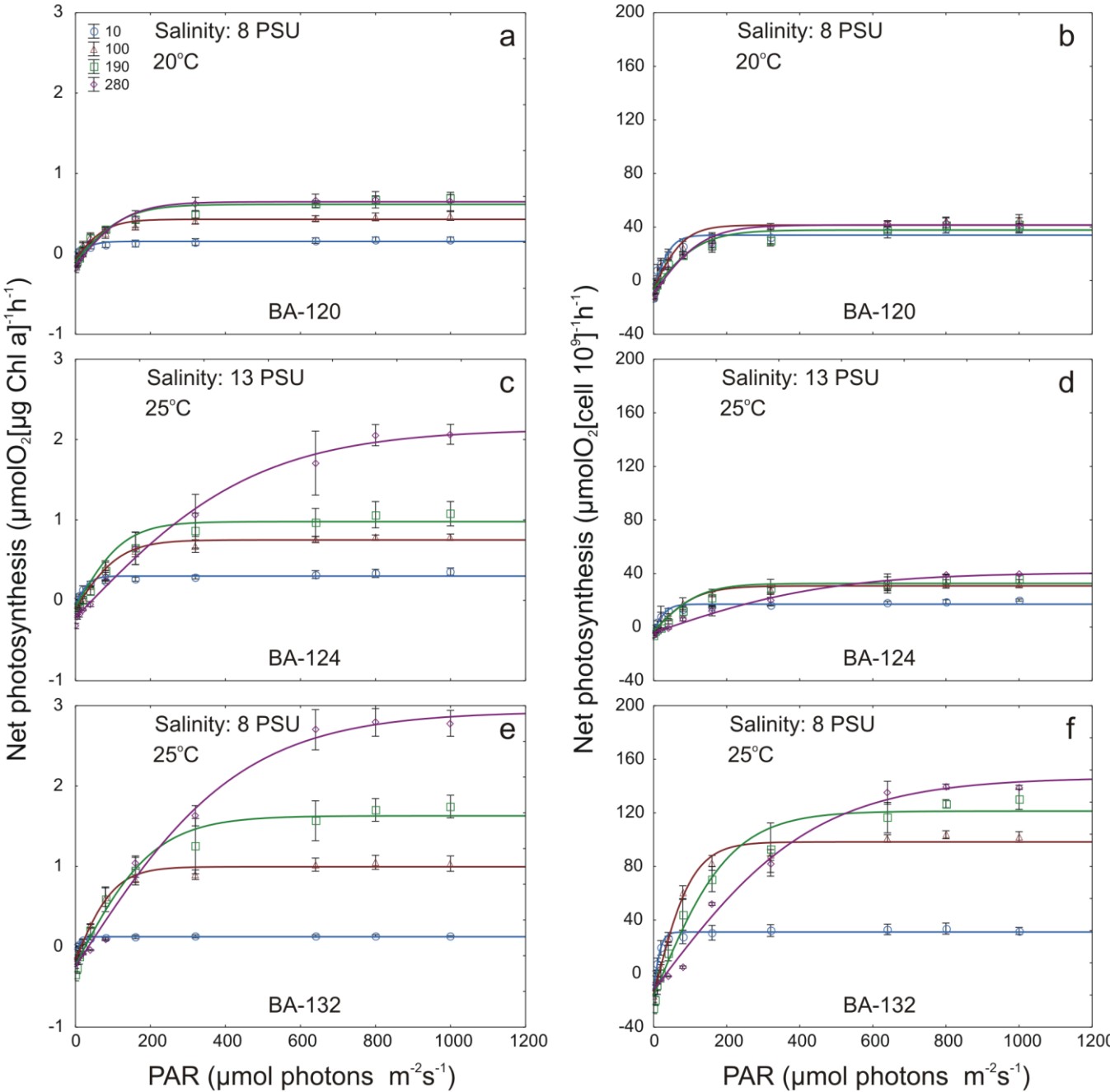

**Figure 6.** Selected Chl *a* - specific and cell-specific (right side and left side panel, respectively) net photosynthetic–light response curves for three *Synechococcus* sp. strains: BA-120 (a, b), BA-124 (c, d) and BA-132 (e, f) strains. Curves present examples of three types of photoacclimation observed for *Synechococcus* sp. and these are as follows: change in number of photosynthesis units (PSU) (a, b), change in size of PSU (c, d) and change in accessory pigments activity (e, f).




**Table 1.** Photoacclimation types (mechanisms) for three *Synechococcus* sp. strains: BA-120, BA-124 and BA-132 at different ecological conditions. OTHER stands for altering of accessory pigments activity or changes in enzymatic reactions; PSUsize stands for the change in PSU sizes; PSUno. stands for the change in PSU number. The symbols of labels indicate the strain for which the mechanism is observed and are as follows: [red] for BA-120, [green] for BA-124 and [brown] for BA-132.

| CONDITIONS | Salinity 3 PSU | Salinity 8 PSU | Salinity 13 PSU | Salinity 18 PSU |
|---|---|---|---|---|
| 10°C | PSUsize [brown] | OTHER [red] PSUsize [green] PSUsize [brown] | PSUsize [red] OTHER [red] OTHER [green] | OTHER [red] PSUsize [green] PSUsize [brown] |
| 15°C | - | PSUsize [green] | OTHER [red] PSUsize [green] OTHER [brown] | PSUsize [brown] |
| 20°C | - | PSUno. [red] OTHER [green] PSUsize | PSUsize (or OTHER) [red] OTHER [green] PSUsize | PSUsize [green] |
| 25°C | OTHER [red] PSUsize [brown] | PSUsize [red] PSUsize [green] OTHER [brown] | PSUsize [red] PSUsize [green] PSUsize [brown] | PSUsize [green] PSUsize [brown] |