# Peer review of "Ecophysiological characteristics of red, green and brown strains of"

_Biogeosciences, 2018_

## Referee Comment (RC1) · C. Callieri (Referee) · 26 Feb 2018

1. Please read with attention the text and the legends and be more precise in the description of the experiments and of the results. 2. It would be interesting to show in the supplementary material the photograph of the cultures together with the absorption spectra and the fluorescence spectra to be sure of the PUB presence in the brown cultures. 3. If I well understood you kept the cultures at the different condition combinations for 2 days for acclimation then you used these cultures as inoculum with an initial number of 106 cells ml-1 and the experiment lasted 1 week and at the end you made

the measurement. In this way you calculate the growth not from a curve with different points but with a line from T0 to T7. I wonder why you did not sampled every day to have a better pattern of what happened in the cultures? 4. Why did you use so low flux (14 $\mu$l min-1) with Accuri C6? 5. I would appreciate to know the tresholds you used to count Pcy and the fluorescences you finally selected. 6. In general, the description of the methods should be improved and more detailed. I do not understand line 122-125 were you declare not to consider the cell number but the cell growth: please explain better this concept.

a revision for the language is necessary

---

## Referee Comment (RC2) · Anonymous Referee #2 · 6 Mar 2018

This article is about the physiological characterisation of three Synechococcus strains the Baltic sea. Overall the experiments seem to be well conducted, although it fails to explain the relevance of such study. The general conclusions should be restricted to the results of the study only. The text can be generally understood, however there are some confusing sentences and paragraphs that perhaps could be improved by proof-reading. - The three strains characterised in this paper presented different pigmentation -it would be useful to know whether those strains are clade representatives (is that information available?), how phylogenetically similar they are or any other reason

why they were chosen for the study (are these bloom-forming strains?). - The authors should be consistent when referring to parameters and strains, for example strains are sometimes mentioned by their name and other by their pigment. - is salinity measured in PSU (practical salinity units)? how is that range (3 to 18) compared to Baltic sea water salinity? - are the temperature and PAR ranges representative of the Baltic sea environment? - relevant bibliography is absent from the introduction (e.g. Flombaum et al. 2013, PNAS, and Six et al. 2007, Genome biol.) - line 43: please include a reference that puts Synechococcus as a major bloom contributor - line 56: Sorokin and Zakuskina (2010) studied the bloom in Comacchio lagoon, it is an overclaim to say it is a phenomenon in Europe - paragraph from line 59 is repetitive and does not give much information, please consider re-phrasing -The methods section should be more specific. For example, how was the media prepared in order to change the salinity? where any of the components in f/2 media replaced by Tropic marine synthetic sea salt or was it added on top of it? What pore-size filters were used? - please state xg rather than rpm (or else specify rotor/centrifuge used) - growth rate has to be measured during exponential growth. The parameters here calculated only report yield and not growth rate. - line 131: please put reference or protocol for Chl a and Car extraction - please change "absorption" for "absorbance" - line 147: it is not clear whether the filter or filtrate was used - The results section describes individual strains, but the figures are difficult to interpret. Please consider reviewing labels and legends. - number of cells and growth should not be used interchangeable - it is not clear what a "positive" or "negative" impact means - The "pigment content" section is not clear, please specify in the methods section - Table 1 is very difficult to interpret -how were those parameters measured? - line 401: what does it mean growth intensity? - line 458: how could these variables be related to the natural conditions in different regions of the Baltic sea? - lines 471-473: unclear, please rephrase or delete

---

## Author Comment (AC2) · 21 Mar 2018

1. This article is about the physiological characterisation of three Synechococcus strains the Baltic sea. Overall the experiments seem to be well conducted, although

it fails to explain the relevance of such study. The general conclusions should be restricted to the results of the study only. The text can be generally understood, however there are some confusing sentences and paragraphs that perhaps could be improved by proofreading.

REPLY: The authors would like to thank Reviewer 2 for the comments and suggestions, and to inform that appropriate corrections have been made in the revised MS. The authors inform that the revised MS is ready. In the new version, a series of Reviewer's comments were addressed and the text was revised again. Due to that, we hope the present MS is satisfactory.

2. The three strains characterised in this paper presented different pigmentation -it would be useful to know whether those strains are clade representatives (is that information available?), how phylogenetically similar they are or any other reason why they were chosen for the study (are these bloom-forming strains?). REPLY: We modified the text accordingly by adding information about Synechococcus sp. clades. We also explained in more detail why we chose these strains in our study. All the aspects are addressed in the revised MS (L42-58, L79-83).

Here we enclose the part of the revised MS Introduction section, which was extended in order to introduce above mentioned information.

[revised manuscript text omitted]

3. The authors should be consistent when referring to parameters and strains, for example strains are sometimes mentioned by their name and other by their pigment. REPLY: We corrected this aspect. From the Results section onwards, the strains are referred by their names and parameters by their symbols.

4. a) is salinity measured in PSU (practical salinity units)? REPLY: Yes, we measured salinity in PSU (L109). We added this unit in whole MS. L109: Salinity of the media was measured in PSU (practical salinity units).

b) how is that range (3 to 18) compared to Baltic sea water salinity? The Baltic Sea horizontal salinity gradient is high and different sub-basins are characterized by different mean salinity values. The gradient decreases North towards. The highest salinity is observed in the Baltic Sea boundary to the North Sea (Skagerrak, around, salinity 30), while the lowest mean salinity is observed in the Baltic northernmost regions (around 3 in Bothnian Basin). The concise information about that was introduced to the MS (L117-119) and more detail information was added in Discussion (L625-632).

L117-119: The synthetic environmental conditions of salinity and T applied in the laboratory are representative for the Baltic Sea area (Feistel et al., 2008; 2009; Siegel and Gerth, 2017).

L625-632: Furthermore, the salinity ranges applied in the experiment are also Baltic's representatives. The Baltic Sea horizontal salinity gradient is high and different sub-basins are characterized by different mean salinity values. The gradient decreases North towards. The highest salinity is observed in the Baltic Sea boundary to the North Sea (Skagerrak, mean salinity ranges between 28.34 and 32.71), while the lowest mean salinity is observed in the Baltic northernmost regions (around 2.35 – 3.96 in Bothnian Basin). These numbers were determined on the basis of climatological data from the Baltic Atlas of Long-Term Inventory and Climatology (Feistel et al., 2008; 2010). Thus, the presented analysis may derive accurate assumptions regarding the regional distribution of Synechococcus sp. strains in the Baltic Sea.

5. are the temperature and PAR ranges representative of the Baltic sea environment? REPLY:

The temperature conditions applied in the laboratory are representative for the Baltic Sea area (Siegel and Gerth, 2017). Regarding PAR, its levels has been generated the

highest possible to be achieved in the laboratory. These values are generally lower than mean PAR intensities being observed in the summertime in the Baltic (Leppärranta and Myrberg, 2009). Moreover, the values of environmental conditions variables (salinity, temperature, PAR) were also specified in certain ranges to make this study comparable with other laboratory cultures experiments available in literature. We added the necessary information in L117-123. Additionally, please note that the annotation regarding the laboratory and natural Baltic ecological conditions was also introduced to the Discussion (L616-632).

L616-632: Due to occurrence of extremes in salinity and other environmental conditions in the Baltic Sea area, the Baltic inhabitants are highly adapted to different regions and often reach their physiological limits (Sjöqvist et al., 2015). The changing environmental conditions the cultures were grown in during the experiments were salinity, T and PAR. Daily mean sea surface temperature (Leppäranta and Myrberg, 2009) presents strongly pronounced annual cycles in the Baltic Sea area. Sea surface temperature (SST) range between about 10 and 20°C may be timed in the Baltic between June and September with some inter-annual changes (Siegel and Gerth, 2017). SSTs reaching and exceeding 20 °C are also observed in the Baltic basin. For instance, according to Siegel and Gerth (2017), SSTs higher than 20 °C were recorded in almost whole Baltic area beyond Danish Straits, Bothnian Bay and northern Bothnian Basin in the warmest week of 2016, in July. According to above, the temperatures, under which the picocyanobacterium cultures were grown in the present study ($10 - 25°$C) can be defined as representative for the Baltic Sea.

6. relevant bibliography is absent from the introduction (e.g. Flombaum et al. 2013, PNAS, and Six et al. 2007, Genome biol.) REPLY: These studies are cited in the current version, where appropriate.

7. line 43: please include a reference that puts Synechococcus as a major bloom contributor REPLY: We added the necessary information in L60 by including the reference to Beardall, 2008.

8. a) line 56: Sorokin and Zakuskina (2010) studied the bloom in Comacchio lagoon, it is an overclaim to say it is a phenomenon in Europe REPLY: We agree that this fragment did not give detail information and did not justified our motivation to conduct the present research. We modified the text accordingly, by removing this statement.

b) paragraph from line 59 is repetitive and does not give much information, please consider re-phrasing Thank you for this comment and drawing our attention to the occurrence of the repetition. We re-phrased the paragraph and deleted the repetition in the text (L79-83 in the revised MS).

L78-83: However, there is still a need to provide more systematic information about these organisms. What is more, the need is amplified by the fact that there are only few research papers on the brown strain of Baltic Synechococcus sp. (Stal et al., 2003; Haverkamp et al., 2008; 2009; Jodłowska and ÅŽliwińska, 2014). This gives limited knowledge of PCY and their life cycle in the Baltic Sea, as brown form also contributes to total pico- and phytoplankton biomass in the area of interest (Stal et al., 2003). The above strengthens the motivation to conduct studies on the brown strain of Synechococcus sp.

9. The methods section should be more specific. For example, how was the media prepared in order to change the salinity? where any of the components in f/2 media replaced by Tropic marine synthetic sea salt or was it added on top of it? What pore-size filters were used? REPLY: We corrected this aspect and added more specific information in Methods section (L107-112, 153-154, 157-159, 171-172).

Here, we cited the chosen parts of the revised manuscript Material and methods section. These parts include the modifications, which were introduced. The modifications are marked in colors: regarding the comments of Reviewer 1 – in blue; regarding the comments of Reviewer 2 – in green.

2.1 Material and culture conditions

Three different phenotypes of picocyanobacteria strains from the genus Synechococcus were examined: BA-120 (red), BA-124 (green), and BA-132 (brown). The cultures preparation was carried out as follows. The Synechococcus sp. strains were isolated from the coastal zone of the Gulf of Gdansk (southern Baltic Sea) and maintained as unialgal cultures in the Culture Collection of Baltic Algae (CCBA) at the Institute of Oceanography, University of Gdańsk, Poland (Latała et al., 2006). The experiments on the 'batch cultures' were carried out in 25 mL glass Erlenmeyer flasks containing sterilized f/2 medium (Guillard, 1975). In order to develop the media, the appropriate amount of Tropic Marine Synthetic Sea Salt was dissolved in distilled water. The final salinity was 3, 8, 13 and 18 PSU, measured with salinometer (inoLab Cond Level 1, Weilheim in Oberbayern, Germany). Salinity of the media was measured in PSU (practical salinity units). The major nutrients, microelements and vitamin concentrations were added according to a method proposed by Guillard (1975) (any of the components in f/2 media were not replaced by Tropic Marine Synthetic Sea Salt). Culture media was prepared with artificial seawater filtered through a 0.45-$\mu$m filters (Macherey-Nagel MN GF-5) using a vacuum pump (600 mbar) and autoclaved. Into 25 mL Erlenmeyer glass flasks, the cells of specific strains were inoculated. The picocyanobacteria cultures were acclimated to the various synthetic environmental conditions for two days. The conditions were the combinations of different values of: scalar irradiance in Photosynthetically Active Radiation (PAR) spectrum (10, 100, 190 and 280 $\mu$mol photons m–2 s–1), temperature (T) (10, 15, 20 and 25°C), and salinity (3, 8, 13 and 18 PSU). Values of quantities representing each environmental condition were applied at the fixed intervals, i.e.: PAR, interval 90; T, interval 5; salinity, interval 5. The synthetic environmental conditions of salinity and T applied in the laboratory are representative for the Baltic Sea area (Feistelet al., 2008; 2009; Siegel and Gerth, 2017). Regarding PAR, its levels has been generated the highest possible to be achieved in the laboratory. These values are generally lower than mean PAR intensities being observed in the summertime in the Baltic (Leppäranta and Myrberg, 2009). Moreover, the values of environmental conditions variables (salinity, temperature, PAR) were also specified in certain ranges

[revised manuscript text omitted]

10. please state xg rather than rpm (or else specify rotor/centrifuge used) REPLY: The rotor unit has been changed in revised MS (L157-159).

L157–159: To remove cell debris and filter out the particles, the extracts were centrifuged at 10,000 rpm (8496 $\times$ g) for 5 min (Sigma 2-16P, Osterode am Harz, Germany).

11. growth rate has to be measured during exponential growth. The parameters here calculated only report yield and not growth rate. REPLY: That is right. We calculated the growth rate basing on the abundance difference between the seventh and first days of the experiment (line from T0 to T7). The rationale for that was our intention to focus on the population yield, as the first idea. However, in the context of the present MS we agree that the modification of our approach is needed to be done. Concerning above, we modified this aspect and in the revised MS not a growth rate but the change in number of picocyanobacteria cells within a course of a week is described. We are grateful the Reviewer for this comment.

12. line 131: please put reference or protocol for Chl a and Car extraction REPLY: The reference has been added in the revised MS (L153). L152-154: The concentration of

photosynthetic pigments of analyzed picocyanobacteria was measured by the spectrophotometric method (Strickland and Parsons, 1972). The analysis of mL-specific (pigment content per mL) and cell-specific (pigment content per cell) pigmentation was conducted.

13. please change "absorption" for "absorbance" REPLY: We introduced this change (L159, L163).

L159-163: The absorbance of pigments was estimated on the basis of Beckman spectrophotometer UV-VIS DU 530 measurements at specific wavelengths (750, 665 and 480 nm), using 1 cm quartz cuvette. Pigment concentration was calculated according to Strickland and Parsons (1972). The following formulas have been used: Chl a ($\mu$g mL–1) = 11.236(A665-A750)Va/Vb, Car ($\mu$g mL–1) = 4(A480-A750)Va/Vb, where: Va - extract volume (in this study 5 mL), Vb - sample volume (in this study 4 mL), and Ax - absorbance estimated at wavelength x in a 1-cm cuvette.

14. line 147: it is not clear whether the filter or filtrate was used REPLY: The information has been specified in the revised MS (L171-172).

L171-172: Samples were filtered through 13-mm glass fiber filters (Whatman GF/C, pore size = 1.2 $\mu$m). Before measurement, the filtered sample was kept in the dark for 10 min.

15. The results section describes individual strains, but the figures are difficult to interpret. Please consider reviewing labels and legends. REPLY: The modifications were introduced in the revised MS.

16. number of cells and growth should not be used interchangeable REPLY: Thank you for drawing our attention to this. We corrected this aspect in the revised MS.

17. it is not clear what a "positive" or "negative" impact means REPLY: These sentences were rewritten to be more precise. Moreover, the appropriate brief explanation was introduced to the revised MS (L247-249). The explanation is as follows:

Positive impact means the increasing (positive) dependency, whilst negative impact means decreasing (negative) dependency between the independent and dependent variable, e.g.: between T and abundance.

18. The "pigment content" section is not clear, please specify in the methods section
REPLY: The "pigment content" section was re-phrased. More specific information was also introduced to the Method section (L152-154).

2.3 Determination of the pigments content

The concentration of photosynthetic pigments of analyzed picocyanobacteria was measured by the spectrophotometric method (Strickland and Parsons, 1972). The analysis of mL-specific (pigment content per mL) and cell-specific (pigment content per cell) pigmentation was conducted. After seven days of incubation, 4 mL of culture was filtered in order to separate the picocyanobacteria cells from the medium. Chl a and carotenoids (Car) were extracted from the picocyanobacteria cells with cold 90% acetone (5 mL). To improve extraction, the cells were disintegrated for two minutes by ultrasonication. Then, the test-tube with the extract was held in the dark for three hours at -60°C. To remove cell debris and filter out the particles, the extracts were centrifuged at 10,000 rpm (8496 × g) for 5 min (Sigma 2-16P, Osterode am Harz, Germany). The absorbance of pigments was estimated on the basis of Beckman spectrophotometer UV-VIS DU 530 measurements at specific wavelengths (750, 665 and 480 nm), using 1 cm quartz cuvette. Pigment concentration was calculated according to Strickland and Parsons (1972). The following formulas have been used: Chl a ($\mu$g mL–1) = 11.236(A665-A750)Va/Vb, Car ($\mu$g mL–1) = 4(A480-A750)Va/Vb, where: Va - extract volume (in this study 5 mL), Vb - sample volume (in this study 4 mL), and Ax - absorbance estimated at wavelength x in a 1-cm cuvette.

19. Table 1 is very difficult to interpret. How were those parameters measured? REPLY: We corrected this aspect and added more specific information in the Result (L401-524 and L527-531) and Discussion (L596-602 and L609-610) sections. The description

of photosynthesis parameters measurement is provided in section 2.5 of the MS, i.e.: Measurements of photosynthesis rate.

Here we enclose only a part of a description in the revised Results section, where the modifications were introduced:

The analysis of photosynthesis characteristics enabled examining and defining the photoacclimation process of all three strains of Synechococcus sp. This was done on the basis of the photosynthetic parameters (Figs. S4-S6) and Photosynthesis-Irradiance (P-E) curves (exemplification shown in Fig. 6). The curves were plotted on the basis of laboratory results (Clark oxygen electrode measurements) using the equation of Jassby and Platt (1976). According to a photoacclimation model description (Prezelin, 1981; Prezelin and Sweeney, 1979; Ramus, 1981; Richardson et al., 1983; Pniewski et al., 2016), the results of the present study indicated changes in Photosynthetic Units (PSU) sizes as the photoacclimation mechanism, which occurred most frequently (Table 1).

and in the revised Discussion section:

The results showed that T, PAR and salinity influenced the photosynthesis parameters only to a certain degree. There were many not statistically significant multiple comparisons pointed by post hoc tests. However, it was found that generally, in cell-specific estimations, elevated PAR had a negative effect on $\alpha$ and PAR increase influenced the respiration negatively. For each of the studied strains of Synechococcus sp., the highest $\alpha$ and the lowest Rd were noted for the cells grown under the lowest PAR (10 $\mu$mol photons m–2 s–1). On the other hand, the highest values of Pm were noted at the highest PAR. It pointed to inability for the cells incubated in low PAR conditions to be as effective in photosynthesis as the cells grown under high irradiances. According to our results, on the basis of P-E curves, three types of photoacclimation mechanisms of Synechoccocus sp. were observed: change in PSU size, change in PSU number and altering accessory pigments activity and changes in enzymatic reactions. This was

a striking observation because in the literature results predominantly derive the two first aforementioned types of recognition (Stal et al., 2003; Jodłowska and ÅŽliwińska, 2014). The present study showed that changes in PSU size occur most frequently (Table 1). The second, ranked by frequency of occurrence, was the altering of accessory pigment activity. PSU number changes in Synechoccocus sp. rarely occurred, which is consistent with literature (Jodłowska and ÅŽliwińska, 2014). Moreover, in our study, photoacclimation mechanisms occurred less frequently in the scenarios with salinity 3 PSU. The changes of photosynthesis parameters (Pm, $\alpha$, Rd) under different environmental conditions explains the occurrence of different photoacclimation mechanisms. According to our results, Synechococcus strains present different ecophysiological characteristics, however, they all demonstrate their tolerance to elevated PAR (for BA-120 to a certain degree) and T levels and could have effectively acclimated to varied water conditions. These strains were able to change the composition of photosynthetic pigments in order to use light quanta better. The ability of Synechococcus to sustain their growth in low light conditions and their low photo-inhibition in exposure to high light intensities could give picocyanobacteria an advantage in optically changeable waters (Jasser, 2006).

Thereby, we ensure that the Results section has been extended by introducing the detail description of photosynthesis parameters.

20. line 401: what does it mean growth intensity? REPLY: These sentences were re-phrased.

21. line 458: how could these variables be related to the natural conditions in different regions of the Baltic sea? REPLY: We understand that the Reviewer 2 is asking about how the environmental variables applied in laboratory are relatd to the natural conditions in different regions of the Baltic Sea. We re-phrased the paragraph loacated originally between L458 and L466 (see L616-632).

L616-639: Due to occurrence of extremes in salinity and other environmental conditions in the Baltic Sea area, the Baltic inhabitants are highly adapted to different regions and often reach their physiological limits (Sjöqvist et al., 2015). The changing environmental conditions the cultures were grown in during the experiments were salinity, T and PAR. Daily mean sea surface temperature (Leppäranta and Myrberg, 2009) presents strongly pronounced annual cycles in the Baltic Sea area. Sea surface temperature (SST) range between about 10 and 20°C may be timed in the Baltic between June and September with some inter-annual changes (Siegel and Gerth, 2017). SSTs reaching and exceeding 20 °C are also observed in the Baltic basin. For instance, according to Siegel and Gerth (2017), SSTs higher than 20 °C were recorded in almost whole Baltic area beyond Danish Straits, Bothnian Bay and northern Bothnian Basin in the warmest week of 2016, in July. According to above, the temperatures, under which the picocyanobacterium cultures were grown in the present study (10 − 25°C) can be defined as representative for the Baltic Sea. Furthermore, the salinity ranges applied in the experiment are also Baltic's representatives. The Baltic Sea horizontal salinity gradient is high and different sub-basins are characterized by different mean salinity values. The gradient decreases North towards. The highest salinity is observed in the Baltic Sea boundary to the North Sea (Skagerrak, mean salinity ranges between 28.34 and 32.71), while the lowest mean salinity is observed in the Baltic northernmost regions (around 2.35 − 3.96 in Bothnian Basin). These numbers were determined on the basis of climatological data from the Baltic Atlas of Long-Term Inventory and Climatology (Feistel et al., 2008; 2010). Thus, the presented analysis may derive accurate assumptions regarding the regional distribution of Synechococcus sp. strains in the Baltic Sea. For instance, a salinity horizontal gradient can be one of the factors determining the abundance of a certain strain in the basin. More saline waters are most preferred by BA-132. On that basis, one can assume the concentration of this strain will be higher near the Baltic Sea entrance (Danish Straits) than in Bothnian Bay. Additionally, it was observed that despite elevated PAR conditions being more suitable for BA-124 and BA-132 to grow intensively, all analyzed strains were able to survive and grow in low PAR conditions. This is consistent with other previously published Baltic

studies (Stal et al., 2003; Jodłowska and ÅŽliwińska, 2014) stating that this is caused by phycobilisomes, which are structural components of picocyanobacteria PSII photosystem. The presence of PCY cells throughout the whole euphotic water column was also reported in limnological studies (Becker et al., 2004, Callieri, 2007).

22. lines 471-473: unclear, please rephrase or delete REPLY: We re-phrased the fragment in the revised MS (L645-647).

L645-647: The present paper derives the new knowledge on the BA-132 responses to changing ecological conditions. What is more, the study places BA-132 among the other Synechococcus sp. strains and compares their ecophysiology pointing to significant differences between these organisms.

---

## Author Comment (AC3) · 21 Mar 2018

C. Callieri (Referee #1) c.callieri@ise.cnr.it

REPLY: The authors would like to thank professor Cristiana Callieri for her comments and suggestions, and to inform that appropriate modifications have been made in the

revised MS. What is more, the authors inform that the revised MS is ready.

In the revised version, we improved the Material and methods section and introduced more details there. We revised the whole MS and added a photograph of the cultures together with the absorption spectra and the scatter plot of the orange vs. red fluorescence. This is a new Figure in supplementary material (Fig. S1, which we enclose as a part of the Supplement in the interactive response). We hope the revised version will be satisfactory.

1. Please read with attention the text and the legends and be more precise in the description of the experiments and of the results. REPLY: We read the whole MS carefully and afterwards introduced some modifications (in blue in the revised MS). The results are described more precisely now.

2. It would be interesting to show in the supplementary material the photograph of the cultures together with the absorption spectra and the fluorescence spectra to be sure of the PUB presence in the brown cultures. REPLY: The new figure (Figure S1) was added in Supplementary material.

Figure S1: Left-side top panel (A, B, C) – light microscope photographs of three Synechococcus sp. strains (scale bar = 10 $\mu$m) along with the photographs of the cultures in 25-mL glass Erlenmeyer flasks; right-side top panel – scatter plots of orange fluorescence vs. red fluorescence analyzed using a BD Accuri™ C6 Plus flow cytometer and bottom panel (D) – PAR absorption spectra obtained for the mixture of phycobilin pigments for each Synechococcus sp. strain

3. If I well understood you kept the cultures at the different condition combinations for 2 days for acclimation then you used these cultures as inoculum with an initial number of 106 cells ml-1 and the experiment lasted 1 week and at the end you made the measurement. In this way you calculate the growth not from a curve with different points but with a line from T0 to T7. I wonder why you did not sampled every day to have a better pattern of what happened in the cultures? REPLY: That is right. We

calculated the growth rate basing on the abundance difference between the seventh and first day of the experiment (line from T0 to T7). The rationale for that was our intention to focus on the population yield, as the first idea. However, in the context of the present MS we agree that the modification of our approach is needed to be done. Concerning above, we modified this aspect and in the revised MS not a growth rate but the change in number of picocyanobacteria cells within a course of a week is described. We thank the Reviewer for this comment. We ensure that in further work every time we want to analyze the growth rate itself we will use a curve with everyday measurements.

4. Why did you use so low flux (14 $\mu$L min–1) with Accuri C6? REPLY: Selection of this flow rate was based on previous introductory experiments on determining the most relevant effectiveness. We used the procedure proposed and described by ÅŽliwińska-Wilczewska et al. (2018b).

In the revised MS, we added the sentence (L137-138): Selection of the flow rate was based on previous introductory experiments to determine the most relevant effectiveness.

5. I would appreciate to know the tresholds you used to count Pcy and the fluorescences you finally selected. REPLY: In the revised MS, we added the sentences (L138-142): Choosing an adequate discriminator and thresholds plays a key role in recording the cells correctly. The most reasonable solution to record chlorophyll fluorescing cyanobacteria and microalgae is to choose the red fluorescence as the discriminator (Fig. S1) and to select a high threshold, enough to eliminate optical and electronic noise (Marie et al., 2005). Concerning this, the discriminator was set on the red (chlorophyll) fluorescence with a standard threshold of 80,000 on FSC-H.

6. In general, the description of the methods should be improved and more detailed. I do not understand line 122-125 were you declare not to consider the cell number but the cell growth: please explain better this concept. REPLY: In the revised MS,

we improved and introduced more details in Material and methods section (L116-117, L136-142, L147-148, L152-154, 157-159, L171-172).

Here, we cited the chosen parts of the revised manuscript Material and methods section. These parts include the modifications, which we introduced.

2.1 Material and culture conditions

Three different phenotypes of picocyanobacteria strains from the genus Synechococcus were examined: BA-120 (red), BA-124 (green), and BA-132 (brown). The cultures preparation was carried out as follows. The Synechococcus sp. strains were isolated from the coastal zone of the Gulf of Gdansk (southern Baltic Sea) and maintained as unialgal cultures in the Culture Collection of Baltic Algae (CCBA) at the Institute of Oceanography, University of Gdańsk, Poland (Latała et al., 2006). The experiments on the 'batch cultures' were carried out in 25 mL glass Erlenmeyer flasks containing sterilized f/2 medium (Guillard, 1975). In order to develop the media, the appropriate amount of Tropic Marine Synthetic Sea Salt was dissolved in distilled water. The final salinity was 3, 8, 13 and 18 PSU, measured with salinometer (inoLab Cond Level 1, Weilheim in Oberbayern, Germany). Salinity of the media was measured in PSU (practical salinity units). The major nutrients, microelements and vitamin concentrations were added according to a method proposed by Guillard (1975) (any of the components in f/2 media were not replaced by Tropic Marine Synthetic Sea Salt). Culture media was prepared with artificial seawater filtered through a 0.45-$\mu$m filters (Macherey-Nagel MN GF-5) using a vacuum pump (600 mbar) and autoclaved. Into 25 mL Erlenmeyer glass flasks, the cells of specific strains were inoculated. The picocyanobacteria cultures were acclimated to the various synthetic environmental conditions for two days. The conditions were the combinations of different values of: scalar irradiance in Photosynthetically Active Radiation (PAR) spectrum (10, 100, 190 and 280 $\mu$mol photons m–2 s–1), temperature (T) (10, 15, 20 and 25°C), and salinity (3, 8, 13 and 18 PSU). Values of quantities representing each environmental condition were applied at the fixed intervals, i.e.: PAR, interval 90; T, interval 5; salinity, interval 5. The synthetic environmental

[revised manuscript text omitted]

7. A revision for the language is necessary REPLY: The MS was checked and corrected by the professional Proof Reading Service company before submitting, however, the text has been revised again, considering Reviewer's suggestion.

Please also note the supplement to this comment:
https://www.biogeosciences-discuss.net/bg-2018-19/bg-2018-19-AC3-supplement.pdf

---

## Author Response (AR1)

**REPLY:**
The authors would like to thank professor Cristiana Callieri for her comments and suggestions, and to inform that appropriate modifications have been made in the revised MS.

In the revised version, we improved the Material and methods section and introduced more details there. We revised the whole MS and added a photograph of the cultures together with the absorption spectra and the scatter plot of the orange vs. red fluorescence. This is a new Figure in supplementary material (Fig. S1). We hope the revised version will be satisfactory. All the modifications in the MS are marked in blue color.

1. Please read with attention the text and the legends and be more precise in the description of the experiments and of the results.
**REPLY:**
We read the whole MS carefully and afterwards introduced some modifications (in blue). The results are described more precisely now.

2. It would be interesting to show in the supplementary material the photograph of the cultures together with the absorption spectra and the fluorescence spectra to be sure of the PUB presence in the brown cultures.
**REPLY:**
The new figure (Figure S1) was added in Supplementary material.

3. If I well understood you kept the cultures at the different condition combinations for 2 days for acclimation then you used these cultures as inoculum with an initial number of $10^6$ cells ml$^{-1}$ and the experiment lasted 1 week and at the end you made the measurement. In this way you calculate the growth not from a curve with different points but with a line from T0 to T7. I wonder why you did not sampled every day to have a better pattern of what happened in the cultures?
**REPLY:**
That is right. We calculated the growth rate basing on the abundance difference between the seventh and first day of the experiment (line from T0 to T7). The rationale for that was our intention to focus on the population yield, as the first idea. However, in the context of the present MS we agree that the modification of our approach is needed to be done. Concerning above, we modified this aspect and in the revised MS not a growth rate but the change in number of picocyanobacteria cells within a course of a week is described.
We thank the Reviewer for this comment. We ensure that in further work every time we want to analyze the growth rate itself we will use a curve with everyday measurements.

4. Why did you use so low flux (14 µL min$^{-1}$) with Accuri C6?
**REPLY:**

Selection of this flow rate was based on previous introductory experiments on determining the most relevant effectiveness. We used the procedure proposed and described by Śliwińska-Wilczewska et al. (2018b).

In the revised MS, we added the sentence (L137-138):
*Selection of the flow rate was based on previous introductory experiments to determine the most relevant effectiveness.*

5. I would appreciate to know the tresholds you used to count Pcy and the fluorescences you finally selected.
**REPLY:**
In the revised MS, we added the sentences (L138-142):
*Choosing an adequate discriminator and thresholds plays a key role in recording the cells correctly. The most reasonable solution to record chlorophyll fluorescing cyanobacteria and microalgae is to choose the red fluorescence as the discriminator (Fig. S1) and to select a high threshold, enough to eliminate optical and electronic noise (Marie et al., 2005). Concerning this, the discriminator was set on the red (chlorophyll) fluorescence with a standard threshold of 80,000 on FSC-H.*

6. In general, the description of the methods should be improved and more detailed. I do not understand line 122-125 were you declare not to consider the cell number but the cell growth: please explain better this concept.
**REPLY:**
In the revised MS, we improved and introduced more details in Material and methods section (L116-117, L136-142, L147-148, L152-154, 157-159, L171-172).

We also removed the sentences from L122-125:
*The growth rate and cells concentration are different parameters but both lead the researcher to the same conclusions on the growth characteristics. In this paper, the growth rates were analyzed abandoning the separate study on the cell concentrations themselves.*

7. A revision for the language is necessary
**REPLY:**
The MS was checked and corrected by the professional Proof Reading Service company before submitting, however, the text has been revised again, considering Reviewer's suggestion.
1. This article is about the physiological characterisation of three *Synechococcus* strains the Baltic sea. Overall the experiments seem to be well conducted, although it fails to explain the relevance of such study. The general conclusions should be restricted to the results of the study only. The text can be generally understood, however there are some confusing sentences and paragraphs that perhaps could be improved by proofreading.

**REPLY:**
The authors would like to thank Reviewer 2 for the comments and suggestions, and to inform that appropriate corrections have been made in the revised MS. All the modifications in the MS are marked in green color. In the new version, a series of Reviewer's comments were addressed and the text was revised again. Due to that, we hope the present MS is satisfactory.

2. The three strains characterised in this paper presented different pigmentation -it would be useful to know whether those strains are clade representatives (is that information available?), how phylogenetically similar they are or any other reason why they were chosen for the study (are these bloom-forming strains?).
**REPLY:**
We modified the text accordingly by adding information about *Synechococcus* sp. clades. We also explained in more detail why we chose these strains in our study. All the aspects are addressed in the revised MS (L42-58, L79-83).

3. The authors should be consistent when referring to parameters and strains, for example strains are sometimes mentioned by their name and other by their pigment.
**REPLY:**
We corrected this aspect.

4. a) is salinity measured in PSU (practical salinity units)?
**REPLY:**
Yes, we measured salinity in PSU (L109). We added this unit in whole MS.

b) how is that range (3 to 18) compared to Baltic sea water salinity?
The Baltic Sea horizontal salinity gradient is high and different sub-basins are characterized by different mean salinity values. The gradient decreases North towards. The highest salinity is observed in the Baltic Sea boundary to the North Sea (Skagerrak, around, salinity 30), while the lowest mean salinity is observed in the Baltic northernmost regions (around 3 in Bothnian Basin).
The concise information about that was introduced to the MS (L117-119) and more detail information was added in Discussion (L625-630).

5. are the temperature and PAR ranges representative of the Baltic sea environment?
**REPLY:**
We added the necessary information in L117-123.

The temperature conditions applied in the laboratory are representative for the Baltic Sea area (Siegel and Gerth, 2017). Regarding PAR, its levels has been generated the highest possible to be achieved in the laboratory. These values are generally lower than mean PAR intensities being observed in the summertime in the Baltic (Leppärranta and Myrberg, 2009). Moreover, the values of environmental conditions variables (salinity, temperature, PAR) were also specified in certain ranges to make this study comparable with other laboratory cultures experiments available in literature.

Additionally, please note that the annotation regarding the laboratory and natural Baltic ecological conditions was also introduced to the Discussion (L616-632).

6. relevant bibliography is absent from the introduction (e.g. Flombaum et al. 2013, PNAS, and Six et al. 2007, Genome biol.)
**REPLY:**
These studies are cited in the current version, where appropriate (L34; L45).

7. line 43: please include a reference that puts *Synechococcus* as a major bloom contributor
**REPLY:**
We added the necessary information in L60.

8. a) line 56: Sorokin and Zakuskina (2010) studied the bloom in Comacchio lagoon, it is an overclaim to say it is a phenomenon in Europe
**REPLY:**
We agree that this fragment did not give detail information and did not justified our motivation to conduct the present research.
We modified the text accordingly, by removing this statement.

b) paragraph from line 59 is repetitive and does not give much information, please consider re-phrasing
Thank you for this comment and drawing our attention to the occurrence of the repetition.
We re-phrased the paragraph and deleted the repetition in the text (L79-83 in the revised MS).

9. The methods section should be more specific. For example, how was the media prepared in order to change the salinity? where any of the components in f/2 media replaced by Tropic marine synthetic sea salt or was it added on top of it? What pore-size filters were used?
**REPLY:**
We corrected this aspect and added more specific information in Methods section (L107-112, 153-154, 157-159, 171-172).

10. please state xg rather than rpm (or else specify rotor/centrifuge used)
**REPLY:**
The rotor unit has been changed in revised MS (L157-159).

11. growth rate has to be measured during exponential growth. The parameters here calculated only report yield and not growth rate.
**REPLY:**
That is right. We calculated the growth rate basing on the abundance difference between the seventh and first days of the experiment (line from T0 to T7). The rationale for that was our intention to focus on the population yield, as the first idea. However, in the context of the present MS we agree that the modification of our approach is needed to be done. Concerning above, we modified this

aspect and in the revised MS not a growth rate but the change in number of picocyanobacteria cells within a course of a week is described.
We are grateful the Reviewer for this comment.

12. line 131: please put reference or protocol for Chl *a* and Car extraction
**REPLY:**
The reference has been added in the revised MS (L153).

13. please change "absorption" for "absorbance"
**REPLY:**
We addressed this change (L159, L163).

14. line 147: it is not clear whether the filter or filtrate was used
**REPLY:**
The information has been specified in the revised MS (L171-172).

15. The results section describes individual strains, but the figures are difficult to interpret. Please consider reviewing labels and legends.
**REPLY:**
The modifications were introduced in the revised MS.

16. number of cells and growth should not be used interchangeable
**REPLY:**
We corrected this aspect in the revised MS.

17. it is not clear what a "positive" or "negative" impact means
**REPLY:**
These sentences were rewritten to be more precise.
Moreover, the appropriate brief explanation was introduced to the revised MS (L247-249). The explanation is as follows:

Positive impact means the increasing (positive) dependency, whilst negative impact means decreasing (negative) dependency between the independent and dependent variable, e.g.: between T and abundance.

18. The "pigment content" section is not clear, please specify in the methods section
**REPLY:**
The "pigment content" section was re-phrased. More specific information was also introduced to the Method section (L152-154).

19.Table 1 is very difficult to interpret. How were those parameters measured?
**REPLY:**
We corrected this aspect and added more specific information in the Result (L401-524 and L527-531) and Discussion (L596-602 and L609-610) sections.

The description of photosynthesis parameters measurement is provided in section 2.5 of the MS, i.e.: Measurements of photosynthesis rate.

20. line 401: what does it mean growth intensity?
**REPLY:**

These sentences were re-phrased.

21. line 458: how could these variables be related to the natural conditions in different regions of the Baltic sea?
**REPLY:**
We understand that the Reviewer 2 is asking about how the environmental variables applied in laboratory are relatd to the natural conditions in different regions of the Baltic Sea. We re-phrased the paragraph loacated originally between L458 and L466 slightly (see L616-632).

22. lines 471-473: unclear, please rephrase or delete
**REPLY:**
We re-phrased the fragment in the revised MS (L645-647).

---

## Editor Decision (ED1)

Comments on "Ecophysiological characteristics of red, green, and brown strains of the Baltic picocyanobacteirum *Synechococcus* sp. – a laboratory study"

General comments

I read the manuscript and found that the authors arranged the experiment in an elaborate way and the results reported may have some significance in biogeochemistry in the Baltic Sea. However, the authors totally failed to describe what is important and what is the ecological significance. The authors just presented list of outputs in the Result section, which made me fatigue. I believe that this is because the authors were not conscious enough on what is to be clarified in this study. In the Discussion section, the authors make some ecological discussion as if they had just come to this issue for the first time. However, such an issue should have been presented in the Introduction and the authors should have clarified what is the REAL OBJECTIVE. If the authors successfully notice what is the objective, the Result section could have been more arranged with appropriate selection of what is necessary and what is not.

Additionally, wording and phrasing in English were terrible. Actually, I could not catch what the authors meant in some sentences. I felt that many sentences are just literal translations from the authors' mother language. I STRONGLY RECOMMEND the authors to have this manuscript checked and edited by a native English speaker or an editing service.

And the authors should reduce the volume of the manuscript. It is too lengthy and redundant. Probably most of the Results section can be omitted, if the authors notice what is important.

Specific comments

L44 This is confusing and I am afraid that the authors may have misunderstood the pigment of cyanobacteria. PE and PC are apoproteins, while PUB and PEB are phycobilins connected to apoproteins. These are different concepts. The readers may question "What is the phycobilin composition of red and green strains?" or "Both red and brown strains contain PE… What is the difference?"

L84 If this is your overall goal, this journal is not suitable for you. You should submit your manuscript to the journal more oriented to physiology. You should aim at a more ecological issue.

L87 Minute figures for experimental settings should not appear in the Introduction. It appears again in Methods and redundant.

L108 Information on salinity appears again in L116. It is redundant. Generally, this section is too wordy and redundant. Make it clear.

L125 If the authors use halogen lamps only for higher irradiance, there would make difference in the spectrum of the light received among the "scenarios". As the authors know, the wavelength is an important factor for the growth of different strains of *Synechococcus* sp., because the different phycobilin compositions result in different absorption spectra of their photosynthetic antenna. How do the authors explain it?

L147 This sentence should appear first of the paragraph.

L148 Do the authors mean "volume-specific" by "mL-specific"?

L171 Replace ""through" with "onto", because the authors use the particle filtered onto a filter for analysis.

L197 Why did the authors ignore the salinity for independent variable and their confounding effects?

L206 Use tables to show the significance of relationships between the variables. The endless listing of the values does not interest the readers.

L217 The authors just examined for different light intensities, and the setting over 190 is just one setting, 280. Then is it appropriate to use "onward"? The authors cannot tell whether the cell abundance continues to decrease "onward". One biggest concern about that is that it may be inappropriate to use ANOVA to describe such a relationship, because it is based on the assumption of linear relationship between the two variables.

L219 This description is too speculative and far from quantitative. What do you mean by "important"? And on which result is this description based on? This comment points to many other sentences in this section.

L228 Actually, I could not understand what the authors intended to say in these sentences (to L234). Please rearrange them.

L488 From here on, every paragraph is just a repetition of the first one, where some figures may have been replaced. It obscures what is important and which the reader should be reminded of. My question here is only "So what?".

L544 Cite appropriate literatures to support this description.

L573 "Acclimation" means the phenotypic phenomenon that one organism strengthens its ecological fitness by changing gene expression. Do you mean it here or did you intend "adapted"?

L590 Why only in this scenario? Is it a universal phenomenon?

L591 You say "also", but in addition to what?

L617 "the Baltic inhabitants are highly adapted to different regions" This sentence is too abstract.

L620-L625 These should have been placed in Introduction.

L640 This paragraph should have been placed in Introduction.

L644 What is "new information"? How is it related with your results?

L678 "This study shows differences and similarities" This sentence does not give any information. EVERY STUDY shows differences and similarities among different things. How different? Which is how? At which point different? What does it mean?

---

## Author Response (AR2)

**Responses to anonymous Reviewer**

Comments on "Ecophysiological characteristics of red, green, and brown strains of the Baltic picocyanobacterium *Synechococcus* sp. – a laboratory study"

General comments

I read the manuscript and found that the authors arranged the experiment in an elaborate way and the results reported may have some significance in biogeochemistry in the Baltic Sea. However, the authors totally failed to describe what is important and what is the ecological significance. The authors just presented list of outputs in the Result section, which made me fatigue. I believe that this is because the authors were not conscious enough on what is to be clarified in this study. In the Discussion section, the authors make some ecological discussion as if they had just come to this issue for the first time. However, such an issue should have been presented in the Introduction and the authors should have clarified what is the REAL OBJECTIVE. If the authors successfully notice what is the objective, the Result section could have been more arranged with appropriate selection of what is necessary and what is not.

Additionally, wording and phrasing in English were terrible. Actually, I could not catch what the authors meant in some sentences. I felt that many sentences are just literal translations from the authors' mother language. I STRONGLY RECOMMEND the authors to have this manuscript checked and edited by a native English speaker or an editing service.

And the authors should reduce the volume of the manuscript. It is too lengthy and redundant. Probably most of the Results section can be omitted, if the authors notice what is important.

**REPLY:**
The authors would like to thank anonymous Reviewer for the general comment, and hereby they want to inform that appropriate modifications have been introduced to the revised manuscript. Regarding the English writing style, please, note that before the submission, the MS was verified by the professional Proof Reading Service company. Nevertheless, the text has been checked again, following the Reviewer's advice. We hope that the present version is satisfactory. All the modifications in the manuscript are marked in blue color.

Specific comments

L44 This is confusing and I am afraid that the authors may have misunderstood the pigment of cyanobacteria. PE and PC are apoproteins, while PUB and PEB are phycobilins connected to apoproteins. These are different concepts. The readers may question "What is the phycobilin composition of red and green strains?" or "Both red and brown strains contain PE… What is the difference?"
**REPLY:**
Authors are aware of differences between apoproteins and phycobillins. In order to emphasize understanding the fragment pointed by the Reviewer, the Authors decided to rephrase it and provide it with more details. This issue was clarified in the revised manuscript in the way shown below (L: 42-54, in the revised MS)

*Picocyanobacteria of the Synechococcus group span a range of different colors, depending on their pigments composition (Stomp et al., 2007; Haverkamp et al., 2008). Synechococcus sp. ranged by the pigment content are divided into two main groups: strains rich in the pigment phycoerythrin (PE), rendering the representatives a variety of orange, brown, reddish, pink and purple colors, and strains rich in phycocyanin (PC), coloring the organism in various shades of blue-green (Haverkamp et al., 2009). Baltic strains of Synechococcus sp. are classified as three main groups: red and brown strains rich in PE and green strains rich in PC (Mazur-Marzec et al., 2013; Jodłowska and Śliwińska, 2014). The difference between red and brown strains is a proportion of two different bilin pigments known as phycoerythrobilin (PEB) and phycourobilin (PUB), which both bind to the PE apoprotein (Everroad and Wood, 2006; Stomp et al., 2007; Six et al., 2007a, b; Haverkamp et al., 2008; 2009). The three strains of Synechococcus sp.: BA-120 (red), BA-124 (green), and BA-132 (brown) examined in this work (Fig. S1 in Supplement) are different morphotypes representatives. Coexistence of PE and PC-rich picocyanobacteria can be found in waters of intermediate turbidity, such as many freshwater lakes and coastal seas including the Baltic Sea (Andersson et al., 1996; Hajdu et al., 2007; Stomp et al., 2007; Haverkamp et al., 2008; Haverkamp et al., 2009; Mazur-Marzec et al., 2013; Larsson et al., 2014; Paczkowska et al., 2017).*

L84 If this is your overall goal, this journal is not suitable for you. You should submit your manuscript to the journal more oriented to physiology. You should aim at a more ecological issue.
**REPLY:**
According to Authors' best knowledge, the objectives of the paper are suitable for the Biogeosciences' scope. However, in order to highlight the overall goal of this study, the Authors described it in more details in L: 90-93

L87 Minute figures for experimental settings should not appear in the Introduction. It appears again in Methods and redundant.
**REPLY:**
The Introduction section in the paper aim at giving the background information on the issue and motivation to conduct the study. It should also provide for a brief description of what the study is based on. Nevertheless, slight modifications have been introduced, i.e.: the sentence: '*These quantities were as follows: scalar irradiance in Photosynthetically Active Radiation (PAR) spectrum range (10, 100, 190, 280 µmol photons $m^{-2}$ $s^{-1}$), salinity (3, 8, 13, 18 PSU), and temperature (T) (10, 15, 20, 25°C).*' was deleted.

L108 Information on salinity appears again in L116. It is redundant. Generally, this section is too wordy and redundant. Make it clear.
**REPLY:**
We changed "Material and culture conditions" section in order to address Reviewer's suggestions. However, we did not decide to shorten it significantly since the in-depth view of Material and Methods section was demanded by previous Reviewers and the detailed form of it is essential in the MS according to Authors' opinion as well.

L125 If the authors use halogen lamps only for higher irradiance, there would make difference in the spectrum of the light received among the "scenarios". As the authors know, the wavelength is an important factor for the growth of different strains of Synechococcus sp., because the different phycobilin compositions result in different absorption spectra of their photosynthetic antenna. How do the authors explain it?
**REPLY:**
In the study, the Authors used the light sources, which both give PAR spectrum. This was checked in other studies conducted in the Laboratory of Marine Plant Ecology of the Institute of

Oceanography University of Gdańsk. The confirmation is provided in Jodłowska and Latała (2010) and Jodłowska and Śliwińska (2014). These references were added in the text (L: 124-125). What is more, according to the LI-COR manual and technical specification therein, the sensor analyzes the spectrum and if it responds to Photosynthetically Active Radiation (PAR) spectrum, the intensity of PAR is measured. Considering the above, the Authors ensure that the scalar irradiance applied in each experiment was PAR.

L147 This sentence should appear first of the paragraph.
**REPLY:**
The Authors thank for drawing their attention to that. They agree with it. The sentence appears at the beginning of the paragraph in the revised MS.

L148 Do the authors mean "volume-specific" by "mL-specific"?
**REPLY:**
We clarified this aspect (L: 161-162).

L171 Replace ""through" with "onto", because the authors use the particle filtered onto a filter for analysis.
**REPLY:**
We corrected this aspect.

L197 Why did the authors ignore the salinity for independent variable and their confounding effects?
**REPLY:**
The Authors aimed at demonstrating the influence of temperature and PAR on picocyanobacteria physiology and examine how different are potential impacts of these variables on the organisms growing in different mediums of salinity. This is why the independent variables in the statistics were temperature and PAR. What is more, please, note that the Authors did not ignore the influence of salinity on the picocyanobacteria physiology at all. The positive influence of increasing salinity is one of the striking observation derived in this study.
Please, note also that in section *2.6 Statistical analyses* of the original MS, the Authors provided for the explanation of why temperature and light had been the independent variables in the study.

L206 Use tables to show the significance of relationships between the variables. The endless listing of the values does not interest the readers.
**REPLY:**
The Authors tried to rearrange the text and replace the statistics listing with tables, however they did not decide to present this information in the manuscript in this form, finally. This is because the tables are of 11 pages, which, even with the listing deleted completely within the MS text, would extend the MS size a lot. Authors claim that organizing the main statistic characteristics in the way it had been previously done is the best solution. However, the Authors agree with the Reviewer that the listing may impede reading for the reader who is not interested particularly in statistics numbers. Due to that, the Authors decided to write the statistics in italic, which noticeably reduces the impediment; see for instance L: 221-223. Now, it is much easier to omit these fragments by the potential reader who is not interested in ANOVA values.

L217 The authors just examined for different light intensities, and the setting over 190 is just one setting, 280. Then is it appropriate to use "onward"? The authors cannot tell whether the cell abundance continues to decrease "onward". One biggest concern about that is that it may be inappropriate to use ANOVA to describe such a relationship, because it is based on the assumption of linear relationship between the two variables.

**REPLY:**

The Authors analyzed the influence of specific environmental conditions on the picocyanobacteria physiology, whereas the conditions were different variables ranged from-to specific limits. The 'onwards' term goes for values within these limits, i.e.: onwards within the ranges not onwards generally. However, indeed, Authors try to extrapolate their observations beyond the analyzed ranges, which is not inappropriate in scientific concluding.

Regarding ANOVA – the validity of applying this method to the present study is obvious, according to Authors' best knowledge. This was done in many ecophysiological studies before (e.g.: Defew et al., 2004; Jodłowska and Latała, 2010). Furthermore, please note that ANOVA is a statistical method, which enables one to examine the influence of independent variables on dependent one. It has nothing to do with linearity.

L219 This description is too speculative and far from quantitative. What do you mean by "important"? And on which result is this description based? This comment points to many other sentences in this section.

**REPLY:**

The Authors meant that the most important environmental factor influencing BA-120 number of cells was temperature (T). This was pronounced the most within lower temperatures (10 and 15°C), where the change in BA-120 abundance along with PAR increase was barely observed (being plainly visible along with T increase at once) (see Fig 1A, a-d).

The additional explanation was added in the text (L: 228-231).

L228 Actually, I could not understand what the authors intended to say in these sentences (to L234). Please rearrange them.

**REPLY:**

The paragraph has been rephrased (L: 240:249).

L488 From here on, every paragraph is just a repetition of the first one, where some figures may have been replaced. It obscures what is important and which the reader should be reminded of. My question here is only "So what?".

**REPLY:**

This is the results section. Since the study provides for many results, the best way to describe them was to organize them in a similar way. According to that, the paragraphs 'look' similarly but it is only about the appearance itself since the content (merit) of each fragment is different. Providing for so many parameters and their values in different environmental conditions for different picocyanobacteria cultures needs a good method to present them in one paper in a concise way. The Authors believe they chose an appropriate method to do this limpidly not leading the reader to confusion. Regarding to 'So what?' question, the answer is provided in Discussion section.

L544 Cite appropriate literatures to support this description.

**REPLY:**

We added appropriate references (L: 558-559).

L573 "Acclimation" means the phenotypic phenomenon that one organism strengthens its ecological fitness by changing gene expression. Do you mean it here or did you intend "adapted"?

**REPLY:**

We changes this aspect (L: 586).

L590 Why only in this scenario? Is it a universal phenomenon?
**REPLY:**
Yes, this is a universal phenomenon but it does not have to occur always and in every environmental conditions. In this study it was observed that PSU number change occurred in BA-120 cultures grown under 20°C in medium of 8 PSU.

L591 You say "also", but in addition to what?
**REPLY:**
Authors thank for drawing their attention to that. The 'also' was replaced. Presently, the sentence sounds as follows: PAR and T were the main factors also in terms of influencing the changes in Chl *a* fluorescence in three strains of *Synechococcus* sp. (L: 606-608).

L617 "the Baltic inhabitants are highly adapted to different regions" This sentence is too abstract.
**REPLY:**
Authors do not consider this sentence as too abstract. What is more, they cite the appropriate literature to confirm their point of view.

L620-L625 These should have been placed in Introduction.
**REPLY:**
Authors do not agree to move this fragment to Introduction section. This is because the paper is not on the environmental conditions in the Baltic Sea explicitly. According to Authors' opinion, placing the fragment in Introduction could lead to confusion while reading the introductory part. Furthermore, Discussion is the section where the results are analyzed and discussed also regarding the natural conditions in the Baltic. Considering that, Authors hold that the sentences were placed appropriately within the MS. What is more, the previous Reviewer suggested to add more detail description on Baltic representative conditions while writing about the application of derived results to natural environment (which is in Discussion section). Since the Authors try to address all the comments on the manuscript, from both the present and the previous revision, and to follow their own opinions at once, they decided to leave the indicated fragment in the Discussion section, as originally (after first revision).

L640 This paragraph should have been placed in Introduction.
**REPLY:**
According to Authors' opinion, this fragment should not be placed in the Introduction section as this fragment involves the results derived from the study. Authors cannot recall observations before deriving them, which is in results section. Considering that, the Authors hold the place to locate this fragment is discussion.

L644 What is "new information"? How is it related with your results?
**REPLY:**
New information means here new (foreground) knowledge. The brown strain of picocyanobacteria has not been examined in such details so far, which is highlighted in the Introduction (L: 84-88) with appropriate literature citation given. In the present paper, the autecology of this strain was analyzed in details (not only the abundance but also photochemical processes (briefly speaking) performed by this strain). Furthermore, the results derived in this study demonstrate how BA-132 is different from the two other analyzed strains. This was possible to be done only because all the strains were grown in exactly the same synthetic environments, meticulously controlled in the laboratory. Additionally, scientific concluding on the strain distribution, living and surviving in the natural Baltic environment was also conducted (oceanic features of Baltic organism – preference of BA-132 to high salinity conditions).

L678 "This study shows differences and similarities" This sentence does not give any information. EVERY STUDY shows differences and similarities among different things. How different? Which is how? At which point different? What does it mean?

**REPLY:**

The sentence: *This study shows differences and similarities* stated only for an introductory sentence in Conclusion section. This was done because in next lines the Authors gave precise and concise information about the differences and similarities observed between strains. Nevertheless, in order to address the anonymous Reviewer's suggestion, Authors decided to delete the first sentence of the Conclusion section.

[revised manuscript text omitted]
 (*alpha*) at two extreme salinities (3 and 18 PSU) under different PAR and temperature conditions for three *Synechococcus* sp. strains: BA-120 (a-d), BA-124 (e-h) and BA-132 (i-l).

---

## Editor Decision (ED2)

Figure 1: Information rich figure.
The 3 strains show different salinity/temperature/light niches
BA120 peaks at ~190 uE, 25C, with niche volume widest at 3-8 PSU
BA124 peaks at ~250 uE, 25C, limited effect of salniity
BA132 peaks at ~280, 25C, widest niche at 3 PSU

Given the distributions one might have wished that the temperature scale went higher to better delimit upper temperature limits; a suggestion for future studies.

Figure 2: information rich figure.
Chl peaks at low light. different temperature effects across strains.

Figure 3. Interesting.
One wonders why Car cell-1 peaks at low light for most strains.

Fig. 6 nice to include both chl-1 and cell-1 specific O2 rates; clearly different.

Abstract
Line 18: 'realistic', not 'real'

introduction;
Line 51: 'morphotypes'?? perhaps 'pigment types'? 'morpho' implies morphology
Same comment on line 63; is 'pigment' part of 'morphology'; maybe 'phenotype'

Materials & Methods:
"The PCY cultures were adapted to the various synthetic environmental conditions for two days. "
How many rounds of cell division occurred during the 2 days of acclimation?
What was the growth state of the cultures before the inoculation into the treatment condition?
Some combinations might have still been in lag phase after 2 days, depending upon the state of the preculture and the severity of the stress.

"The salinity was 120 controlled by salinometer (inoLab Cond Level 1, Weilheim in Oberbayern, Germany). "
Controlled by? Or verified by? Controlled by implies a chemostat type titrator. I think the authors mean they used a salinometer to measure the salinity, not ongoing adjustment of salinity?

What was the photoperiod, and when in the subjective photoperiod were measures taken?

"In order to achieve the most reliable results, test cultures were grown in three replicas and were incubated for one week at 135 each combination of light, temperature and salinity. "
OK, good.
How does this relate to the earlier statement about 2 days?

Figure 1, and first section of results:
Were all replicates initially inoculated at equal cell densities?

Line 142
The flow cytometry was used to establish the initial number of picocyanobacteria cells and to measure the final cells concentration after the incubation period.

It would be more generally useful to express:
$mu = (\ln(Cells_{final}) - \ln(Cells_{initial}))/elapsed\ time$
This assumes steady exponential growth over the entire time window.

Then, $\ln2/mu$ = apparent generation time.
Then elapsed time/generation time = number of generations achieved under each treatment.
Or, the ratio of $Cells_{final}/Cells_{initial}$, as a direct measure of the fold change in biomass.

Any of these metrics would give better comparability across studies.

Instead doing statistics on the achieved final number of cells, without clear reference to the starting number of cells, is methodologically odd and makes comparisons with other studies difficult.

The masses of statistical comparisons make the results almost unreadable.
I wonder if the authors should lift out the masses of statistical comparisons into tabular or supplement form, and use a much shorter text to descibe the main quantitative and qualitative findings, with reference to a table of statistical tests.

Discussion:
"Carotenoids have a dual role in the cell: to maintain a high 599 capacity for photosynthetic light absorption and to provide protection against photooxidation '
I do not know of any evidence that carotenoids can serve in photosynthetic light absorption in cyanobacteria.
If that is true it needs to be backed by a citation.

best regards, Doug Campbell

---

## Author Response (AR3)

The authors would like to thank Dr Douglas Campbell for his comments and suggestions, and to inform that appropriate modifications have been made in the revised MS. We hope that the present version is satisfactory. All the modifications in the manuscript are marked in blue color.

1. **COMMENT:**

Figure 1: Information rich figure.
The 3 strains show different salinity/temperature/light niches
BA120 peaks at ~190 uE, 25C, with niche volume widest at 3-8 PSU
BA124 peaks at ~250 uE, 25C, limited effect of salniity
BA132 peaks at ~280, 25C, widest niche at 3 PSU

Given the distributions one might have wished that the temperature scale went higher to better delimit upper temperature limits; a suggestion for future studies.

1. **REPLY:**

The Authors are grateful for the comment. The motivation underlying the arrangement for scenarios conditions ranges was the will to reflect the general Baltic conditions, to make the study comparable with other known studies on picocyanobacteria autecology, and to make the best use of the laboratory equipment accessible for the Authors at that moment. The Authors do not want to discard the picocyanobacteria studies and they plan to extend the research in the future.

2. **COMMENT**

Figure 2: information rich figure.
Chl peaks at low light. different temperature effects across strains.

2. **REPLY:**

The Authors added this short information into the text (L: 534-546).

*In this study, the pigment content was generally the highest under the low PAR treatment for all strains. This was a striking observation, however, with some exceptions. For instance, concerning Cars, BA-120 cell-specific Car reached high concentrations in the whole light range for high T. This was pronounced in mediums of moderate and high salinity (8, 13, 18). Moreover, in medium 3, BA-124 demonstrated high cell-specific Car concentration under the highest analyzed light level. The cell-specific Car peaking in the lowest light was pronounced the most in BA-132 cultures. This is consistent with literature (Jodłowska and Latała, 2010). Regarding cell-specific Chl a peaks, they were noticeable in the low PAR range for all strains with no exceptions. The difference between the strains were various effects of temperature. For BA-120 and BA-132 the highest cell-specific Chl a concentrations were estimated in the highest T, while for BA-124 oppositely, i.e. for the lowest T. This has not been observed before (according to Authors' best knowledge, not reported in the literature yet). Moreover, the Car/Chl a ratio increase along the PAR increase was observed. This, together with low pigment contents in under high PAR, is a very interesting observation, which makes the motivation for further studies on Synechococcus sp. stronger. The Authors plan to extend*

*their research on picocyanobacteria in the future (the pigment content composition analysis, proportion of Chl a and carotenoids – Zeaxanthin, β-carotene – and Phycobilins).*

**3. COMMENT**

Figure 3. Interesting.
One wonders why Car cell-1 peaks at low light for most strains.

**3. REPLY:**

The observations are described in Resets section (3.2. Pigment content) provided with details. The pigmentation was generally the highest for all strains in the low light range. Please, referrer to Reply 2. and a new fragment added in the text (L: 534-546).

**4. COMMENT**

Fig. 6 nice to include both chl-1 and cell-1 specific O2 rates; clearly different.

**4. REPLY:**

The Authors would like to thank for this comment. They were very interested in what they would obtain for the *P-E* curve trajectory (demonstrating $O_2$ rates) in Chl-specific and cell-specific domains. The results enabled to conclude on photoadaptation mechanisms. In order to analyze the *P-E* curves, the Authors needed to define Chl *a-* and cell-specific photosynthesis parameters.

**5. COMMENT**

Abstract
Line 18: 'realistic', not 'real'

**5. REPLY:**
We corrected this aspect (L: 18).

**6. COMMENT**

introduction;
Line 51: 'morphotypes'?? perhaps 'pigment types'? 'morpho' implies morphology
Same comment on line 63; is 'pigment' part of 'morphology'; maybe 'phenotype'

**6. REPLY:**
We corrected these aspects (L: 51 and L: 62).

**7. COMMENT**

Materials & Methods:
"The PCY cultures were adapted to the various synthetic environmental conditions for two days. "

How many rounds of cell division occurred during the 2 days of acclimation?
What was the growth state of the cultures before the inoculation into the treatment condition?
Some combinations might have still been in lag phase after 2 days, depending upon the state of the preculture and the severity of the stress.

**7. REPLY:**

The Authors would like to thank for this comment. They did not realize, this fragment of Materials and Methods section could have introduced a confusion or misled the Reader. They introduced slight modifications in the text (L: 136-138).
After acclimation time (2 d), the picocyanobacteria cells served as inoculum for the right test cultures with the initial number of cells equal to $10^6$ cells $mL^{-1}$. The acclimation cultures used for inoculation were isolated from the logarithmic growth phase. Due to that, being of some combinations still in lag phase after 2 days can be excluded. During the acclimation time, cell division rate for the strains was about 1 $day^{-1}$, averagely. It was enough to enable the cells to acclimate to environmental conditions without the risk of stress severity.

**8. COMMENT**

"The salinity was 120 controlled by salinometer (inoLab Cond Level 1, Weilheim in Oberbayern, Germany). "
Controlled by? Or verified by? Controlled by implies a chemostat type titrator. I think the authors mean they used a salinometer to measure the salinity, not ongoing adjustment of salinity?

**8. REPLY:**
The Authors thank Dr. Campbell for drawing their attention to that. The salinity was verified by salinometer, of course. This aspect was clarified in the text (L: 123).

**9. COMMENT**

What was the photoperiod, and when in the subjective photoperiod were measures taken?

**9. REPLY:**

We added the sentences (L: 121-122):

*The strains were incubated under a 16:8 h light:dark cycle. The measurements of all strains were taken when the experiment incubations completed (after full 7 days) at the same time during the light:dark cycle (in the light phase).*

**10. COMMENT**

"In order to achieve the most reliable results, test cultures were grown in three replicas and were incubated for one week at 135 each combination of light, temperature and salinity.                                                                                        "
OK, good.
How does this relate to the earlier statement about 2 days?

**10. REPLY:**

'Test cultures' are right cultures to carry the experiment on them. 2 days referred to acclimation cultures (not test ones). The test cultures grown in three replicas under specific conditions for 7 days and then the measurements were taken.

**11. COMMENT**

Figure 1, and first section of results:
Were all replicates initially inoculated at equal cell densities?

**11. REPLY:**

Yes, the initial cells number, i.e. the culture density just after inoculation, was the same for all replicas.
The Authors added this short information into the text (L: 141).

12. COMMENT

Line 142
The flow cytometry was used to establish the initial number of picocyanobacteria cells and to measure the final cells concentration after the incubation period.

It would be more generally useful to express:
mu =( ln(Cellsfinal) - ln(Cells initial))/elapsed time
This assumes steady exponential growth over the entire time window.

Then, ln2/mu = apparent generation time.
Then elapsed time/generation time = number of generations achieved under each treatment.
Or, the ratio of Cellsfinal/Cellsinitial, as a direct measure of the fold change in biomass.

Any of these metrics would give better comparability across studies.
Instead doing statistics on the achieved final number of cells, without clear reference to the starting number of cells, is methodologically odd and makes comparisons with other studies difficult.

**12. REPLY:**

In the first version of the manuscript the analysis were done on the basis of the growth rate. However, that time, the Reviewer recommended the Authors to change the attitude and to do the analysis on the abundances. This is what the Authors followed. What is more, please note that the study is not without a clear reference to the starting number of cells. The Authors pointed to the starting abundances in the text, precisely.
Nevertheless, thought through the suggestions of Dr. Campbell deeply, the Authors decided to add a fragment in the Material and Method section (L: 143-145), which says:

*Additionally, to broaden the understanding and comparison possibilities, the number of generations (number of generations = elapsed time (t) /doubling time (d)) were demonstrated in supplementary materials (Fig. S1).*

**13. **COMMENT**

The masses of statistical comparisons make the results almost unreadable.
I wonder if the authors should lift out the masses of statistical comparisons into tabular or supplement form, and use a much shorter text to descibe the main quantitative and qualitative findings, with reference to a table of statistical tests.

**13.REPLY:**

The Authors inform they modified the Results section and the tables with statistics were added to the supplementary material (Tables S1 – S11)

**14. **COMMENT**

Discussion:
"Carotenoids have a dual role in the cell: to maintain a high 599 capacity for photosynthetic light absorption and to provide protection against photooxidation '
I do not know of any evidence that carotenoids can serve in photosynthetic light absorption in cyanobacteria.
If that is true it needs to be backed by a citation.

**14. REPLY:**

We added citations.

---

## Author Response (AR4)

Dear Professor Koji Suzuki,

Thank you very much for your email, comments and the assistance during the whole reviewing process. We would like to ensure that the editorial corrections and the suggestion by Professor Douglas Campbell provided in your last report were introduced to the present version of the manuscript. We added also some acknowledgements in the *Acknowledgements* section. We hope that the final paper is satisfactory and fully ready to be published. All modifications in the manuscript are marked in blue.

Kind regards,
Agata Cieszyńska
(on behalf of all Authors)
* * *
- Comment from Professor Douglas Campbell:

The authors addressed my concerns. They still include one line:
L526–52: Carotenoids have a dual role in the cell: to maintain a high capacity for photosynthetic light absorption and to provide protection against photooxidation (Siefermann-Harms, 1987; Kana and Glibert, 1988; Jodłowska and Latała, 2010)

and I do not think it is true for cyanobacteria. The cited references do not show that carotenoids act as photosynthetic antenna, rather they show that carotenoid content varies with light acclimation. In many cyanobacteria some carotenoid content is not even in the thylakoid membrane, and I do not know any evidence that shows coupling of carotenoids to photosynthetic light capture in cyanos.

Reply:
The Authors decided to delete the misleading part of the sentence (presently L526-528: Furthermore, carotenoids play an important role in the cells as they provide protection against photooxidation (Siefermann-Harms, 1987; Kana and Glibert, 1988; Jodłowska and Latała, 2010).

- Editorial comments:

L13: The present study examined …
L14: Remove "was conducted".
L19 and L23: among strains.

Reply:
The corrections introduced.